# Temporal inhibition of chromatin looping and enhancer accessibility during neuronal remodeling

Dahong Chen[1,2], Catherine E. McManus [1,2], Behram Radmanesh[3], Leah H. Matzat[1,3] & Elissa P. Lei [1,2✉]

During development, looping of an enhancer to a promoter is frequently observed in conjunction with temporal and tissue-specific transcriptional activation. The chromatin insulator-associated protein Alan Shepard (Shep) promotes *Drosophila* post-mitotic neuronal remodeling by repressing transcription of master developmental regulators, such as *brain tumor* (*brat*), specifically in maturing neurons. Since insulator proteins can promote looping, we hypothesized that Shep antagonizes *brat* promoter interaction with an as yet unidentified enhancer. Using chromatin conformation capture and reporter assays, we identified two enhancer regions that increase in looping frequency with the *brat* promoter specifically in pupal brains after Shep depletion. The *brat* promoters and enhancers function independently of Shep, ruling out direct repression of these elements. Moreover, ATAC-seq in isolated neurons demonstrates that Shep restricts chromatin accessibility of a key *brat* enhancer as well as other enhancers genome-wide in remodeling pupal but not larval neurons. These enhancers are enriched for chromatin targets of Shep and are located at Shep-inhibited genes, suggesting direct Shep inhibition of enhancer accessibility and gene expression during neuronal remodeling. Our results provide evidence for temporal regulation of chromatin looping and enhancer accessibility during neuronal maturation.

[1] Nuclear Organization and Gene Expression Section, Bethesda, MD, USA. [2] Laboratory of Biochemistry and Genetics, National Institute of Diabetes and Digestive and Kidney Diseases, National Institutes of Health, 9000 Rockville Pike, Bethesda, MD 20892, USA. [3] Laboratory of Cellular and Developmental Biology, National Institute of Diabetes and Digestive and Kidney Diseases, National Institutes of Health, 9000 Rockville Pike, Bethesda, MD, USA. ✉email: leielissa@niddk.nih.gov

The establishment and maintenance of proper chromatin topology have emerged as an essential and conserved feature of exquisitely tuned, developmentally regulated gene expression programs. On the finest scale, looping between an enhancer and a promoter (E–P looping) is frequently observed during or even before the onset of developmentally programmed transcriptional activation[1,2]. In fact, forced E–P looping can result in ectopic activation of gene expression[3,4], suggesting that the E–P looping step itself may serve as a point of either positive or negative regulation during development. One such E–P loop-promoting factor is the Lim domain-containing protein Ldb1, which is expressed during the development of specific tissues and forms a complex with particular transcription factors[5]. Although perturbation of transcription factors may indirectly affect E–P loop formation by affecting transcription[6–8], no such dedicated antagonist of E–P looping has yet been identified. Furthermore, the regulation of chromatin 3D structure during post-mitotic neuronal remodeling has not previously been studied.

Architectural proteins, such as insulator proteins, have been demonstrated to participate in the formation of topologically associating domains and cell type-specific E–P loops. The first tissue-specific regulator of insulator activity to be identified is the *Drosophila* RNA-binding protein Shep, which acts as an insulator antagonist only in the nervous system[9,10]. Shep is required for neuronal remodeling, an essential and conserved process that replaces juvenile neuronal projections with adult-specific projections during the metamorphic transition between larval and pupal development. Shep functions in part by repressing transcription of a key neuronal remodeling inhibitor, *brat*, specifically in pupal neurons[11–13]. Shep associates with the chromatin of *brat* and many other target genes[11], frequently at promoters[10]. We, therefore, speculated that Shep may antagonize *brat* E–P looping in order to repress its transcription during post-mitotic neuronal remodeling.

Here, we identify temporal Shep inhibition of *brat* E–P looping during neuronal remodeling. Importantly, Shep does not repress enhancer or promoter activities ex vivo or in vivo, suggesting that Shep primarily inhibits E–P looping to repress *brat* transcription. Moreover, Shep temporally restricts the accessibility of a key *brat* enhancer and other enhancers genome-wide in pupal neurons that correspond to Shep-inhibited gene expression. We conclude that Shep inhibits chromatin looping and enhancer accessibility to regulate gene expression in a stage-specific manner to facilitate neuronal remodeling.

## Results

### Shep inhibits *brat* promoter looping with proximal genomic regions.

In order to survey Shep-dependent *brat* promoter looping, we performed circularized chromosome conformation capture (4C-seq) in central nervous system-derived BG3 cultured cells. Shep inhibits transcription of all *brat* isoforms in BG3 cells and pupal neurons (Fig. 1A–C), and ChIP-seq of Shep indicates chromatin association within the immediate vicinity of each of the annotated *brat* promoters[11] (Fig. 1A). Since *brat-F* is the dominant isoform in pupal neurons[11] (Fig. 1A), a validated Shep chromatin binding site at the *brat-F* promoter (Fig. S1) was selected as an anchor to generate 4C-seq libraries (Fig. 1A). Upon efficient knockdown of Shep (Fig. 1D), we identified two regions within 30 kb of the anchor that display statistically significantly increased interaction with the *brat-F* promoter (Fig. 1A and E, regions 1 and 3) as well as a third region with decreased interaction frequency (region 2). We hypothesized that these regions could harbor *cis*-regulatory elements, such as enhancers, that loop to the *brat-F* promoter to activate its expression. Furthermore, we speculated that Shep may antagonize these putative E–P looping interactions in a temporally regulated manner.

### Shep inhibits *brat* promoter looping to a neural enhancer.

We observed that all three differentially interacting 4C-seq regions are covered by histone post-translational modifications associated with enhancer activity. Examination of publicly available H3K4me1, H3K27ac, and STARR-seq profiles in BG3 cells[14,15] provided evidence that these regions may function as enhancers to regulate *brat* transcription (Fig. 2A). In order to test this possibility, we cloned each of the three putative enhancer regions juxtaposed to the *brat-F* promoter upstream of a firefly luciferase reporter (Fig. 2B) to assay transcription driven by these constructs or by the *brat-F* promoter alone. We found that only region 1 preceding the *brat-F* promoter is able to increase reporter expression in BG3 relative to control constructs (Fig. 2C), whereas region 1 alone does not support substantial luciferase expression (Fig. S2A, B). Interestingly, none of the constructs are able to enhance *brat-F*-dependent expression when transfected into S2 cells, a non-neural cell type[16] (Figs. 2D and S2C), indicating that region 1 is a cell type-specific enhancer.

We next tested whether Shep affects luciferase expression in this artificial context, in which the region 1 enhancer is juxtaposed to the *brat-F* promoter. We repeated the reporter assays with the *brat-F* promoter alone, juxtaposed to the region 1 enhancer, or the region 1 enhancer alone in control vs. Shep-depleted cells. Importantly, no significant differences were observed in control vs. Shep-depleted cells for any of these constructs, indicating that Shep does not directly affect either *brat-F* promoter or region 1 enhancer activities (Fig. 2E). Since Shep associates with regions at or nearby promoters of other *brat* isoforms as well, we also performed reporter assays to test Shep regulation of *brat-A/E* or *brat-B/C* promoters with or without region 1. While region 1 can activate both promoters, depletion of Shep does not affect activity of either promoter, indicating that each of these elements also functions independently of Shep in this artificial context (Fig. 2F, G). These key results suggest that Shep does not simply act as a transcriptional repressor of either enhancer or promoter activities. Taken together, we conclude that Shep repression of *brat* only occurs in the in vivo context, in which we hypothesize that Shep-mediated antagonism of E–P looping in pupae attenuates *brat-F* expression.

### Shep inhibits neural *brat* E–P looping in a stage-specific manner.

In order to further test this hypothesis, we next asked whether Shep inhibits *brat-F* promoter looping with the region 1 enhancer in vivo in a stage-specific manner. We performed directed 3C using Taqman qPCR to quantify looping between the *brat-F* promoter anchor and the surrounding vicinity, including regions 1–3, in isolated control pupal brains vs. brains harboring neurons depleted of Shep (elav > Dcr-2, shep-RNAi, mCD8::GFP) (Fig. 3A). Consistent with our 4C-seq results in BG3 cells (Fig. 1A), we found increased looping in Shep-depleted pupal brains between the *brat-F* promoter and the region 1 enhancer as well as a flanking region, which we named region 4 (Fig. 3A). In contrast, larval brains show no statistically significant differences for *brat-F* promoter interaction frequencies with regions 1 and 4, with low looping interactions in both control and Shep-depleted flies. Interaction frequencies in both larvae and pupae are also low between the *brat-F* promoter and other sites examined, including regions 2 and 3. We further interrogated looping in pupal brains using the *brat-B/C* promoter as an anchor but found very low levels of looping with regions 1 and 4 relative to looping observed with the *brat-F* promoter (Fig. S3), demonstrating a correlation between low looping frequency and low expression of *brat-B/C*. Moreover, Shep depletion in pupal brains has no effect on looping of regions 1 and 4 with the *brat-B/C* promoter (Fig. S3). Examination of Shep chromatin association in the larval brain revealed that Shep does associate with chromatin at the *brat-F* promoter as well as regions 1 and 4 (Fig. 3A), suggesting direct function at these sites. Given that Shep inhibits *brat* expression in neurons specifically at the

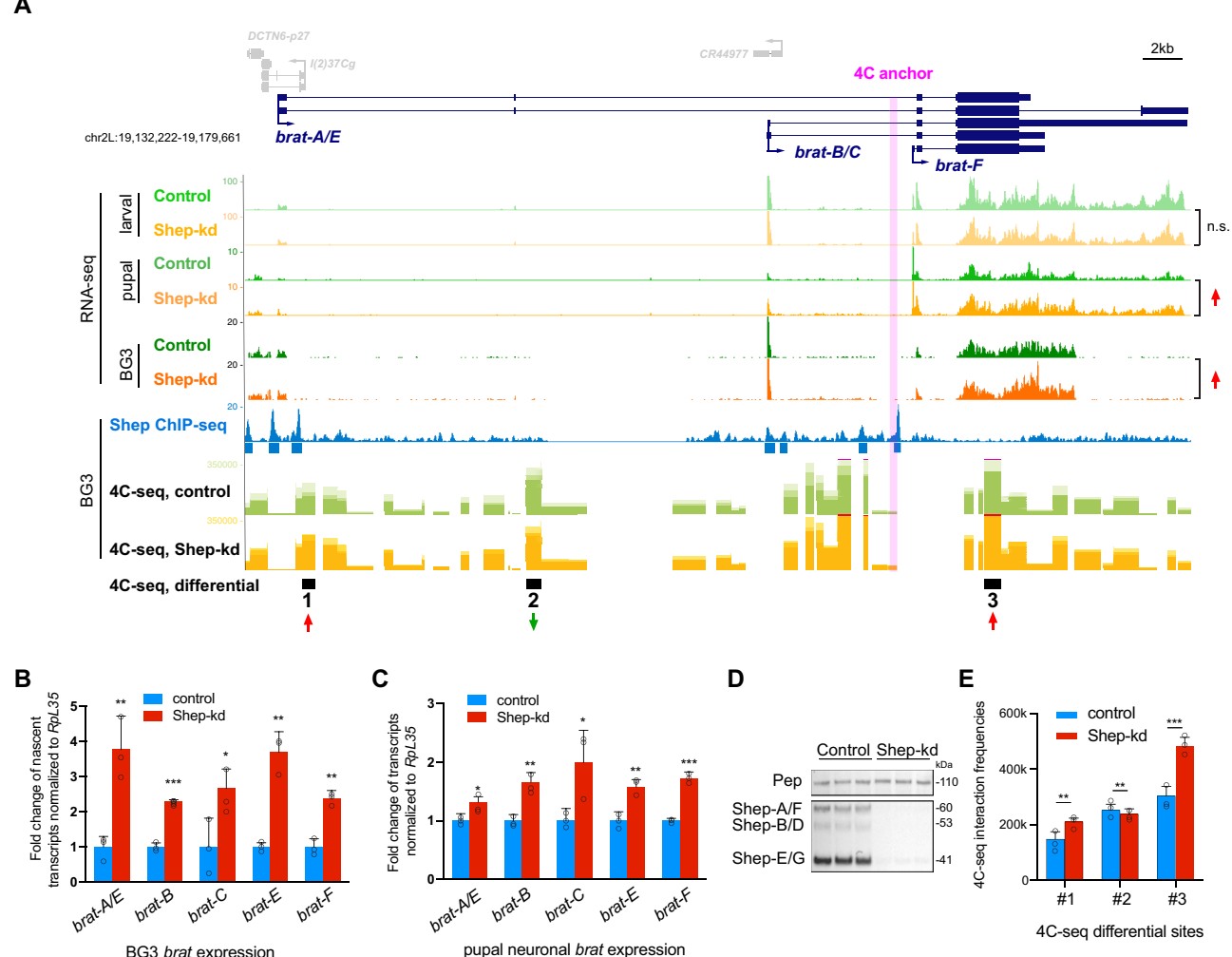

**Fig. 1 Shep inhibits *brat* expression and *brat-F* promoter looping with proximal genomic regions. A** RNA-seq analysis of control vs. Shep-depleted larval neurons, pupal neurons (FDR = 1.6e−3, fold change = 1.4), and BG3 cells (FDR = 5.6e−2, fold change = 1.2) at the *brat* locus (top). Note different scales of RNA-seq tracks between larval and pupal neurons, indicating a dramatic decrease in *brat* expression during neuronal remodeling. ChIP-seq profile of Shep in BG3 cells with called peaks indicated below (middle). BG3 cell 4C-seq using the *brat-F* promoter as an anchor (pink shading) identifies three regions (red and green arrows) with differential (FDR = 7.0e−3, 5.6e−3, and 1.4e−4, respectively; fold change = 1.3, 0.8, and 1.3, respectively) interaction frequencies upon Shep depletion (bottom). Note that only these three regions pass the FDR < 0.01 threshold. **B** Shep depletion in BG3 cells leads to increased transcription of all *brat* isoforms. Nascent RNA was quantified by EU-qPCR using isoform-specific primers. Experiments were performed with $n = 3$ biologically independent samples. Data are presented as mean values + SEM and two-tailed Student's *t* test was performed with *$p < 0.05$, **$p < 0.01$, ***$p < 0.001$. Exact $p$ values are reported in the Source Data. **C** Shep depletion leads to increased steady-state expression of all *brat* isoforms in sorted pupal neurons. Experiments were performed with $n = 3$ biologically independent samples. Data are presented as mean values + SEM and two-tailed Student's *t* test was performed with *$p < 0.05$, **$p < 0.01$, ***$p < 0.001$. Exact $p$ values are reported in the Source Data. **D** Efficient Shep protein depletion achieved by dsRNA treatment in BG3 cells. Pep serves as a loading control for Western blotting. Reproducible efficient depletion was observed in two independent experiments. **E** Graph showing looping frequency measured by 4C-seq in panel A specifically for the three regions displaying differential interaction that pass the statistical significance threshold. Experiments were performed with $n = 3$ biologically independent samples. Data are presented as mean values + SEM, and $p$ values are derived from FourCSeq with **$p < 0.01$, ***$p < 0.001$ (see Methods). Exact $p$ values are reported in the Source Data.

pupal stage, our results provide strong evidence for Shep inhibition specifically of *brat-F* looping with enhancer regions 1 and 4 in a temporal manner.

**Region 4 functions as a pupal enhancer in vivo**. We thus suspected that both regions 1 and 4 may act as enhancers in vivo, so we cloned regions 1 and 4 individually juxtaposed to the *hsp70* promoter upstream of a GFP reporter to test their ability to activate transcription in flies. We inserted these constructs into the *attP40* docking site using *PhiC31* integrase in order to assay GFP expression driven by regions 1 or 4 compared to the *hsp70*

promoter alone. We observed that region 4 strongly activates GFP expression specifically in pupal but not larval brains (Fig. 3D, G). To test its in vivo activity, we CRISPR-deleted the region 4 enhancers in flies and observed reduction of *brat-F* expression in pupal brains (Fig. 3H), indicating a critical function for region 4 as an enhancer of *brat* transcription at this stage of development. On the other hand, region 1 enhances *hsp70* promoter-driven GFP expression only in larval but not pupal brains (Fig. 3C, F). Consistent with the luciferase reporter assays in BG3 cells, enhancer activities of regions 1 and 4 were unchanged in strong loss-of-function *shep* mutant brains (Fig. S4A–D), indicating that Shep does not simply repress enhancer activities. In conclusion,

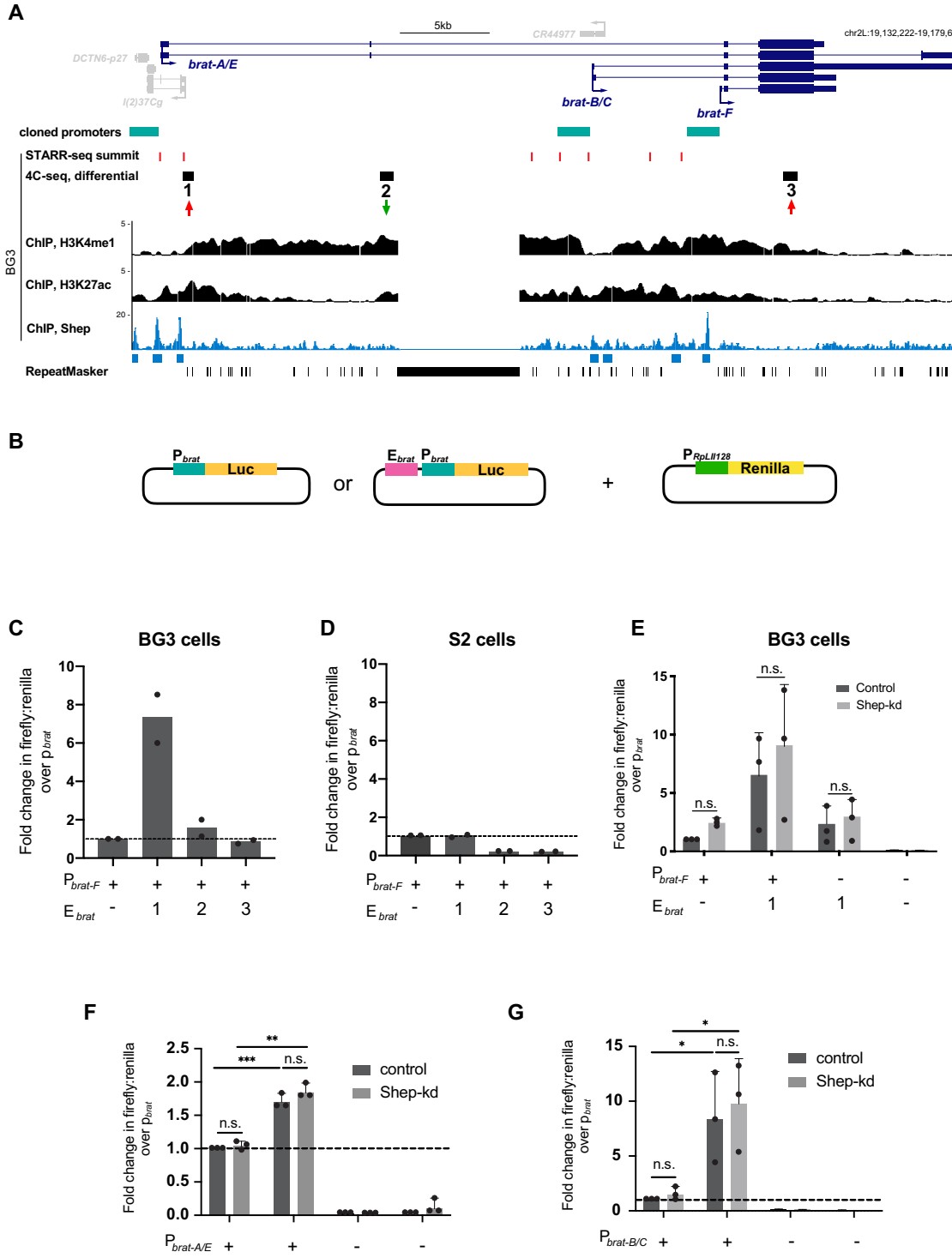

both regions 1 and 4 functions as *brat* enhancers in vivo, in larvae and pupae respectively, and Shep cannot directly repress the activity of either enhancer. These results are consistent with the hypothesis that Shep functions as an antagonist of E–P looping in order to negatively regulate *brat* transcription at the pupal stage.

**Shep inhibits chromatin accessibility of the region 4 enhancer in pupal neurons**. In order to gain insight into the mechanism by which *brat* E–P looping is regulated by Shep, we performed ATAC-

seq to examine chromatin accessibility in control vs. Shep-depleted neurons. To this end, neurons from control larvae or pupae were GFP-labeled (*elav > Dcr-2, mCD8::GFP*) and FACS-isolated for Omni-ATAC-seq analyses[17]. Consistent with an overall decreased expression between larval and pupal stages in control neurons, chromatin accessibility of *brat* overall is also strongly reduced during this developmental transition (Fig. 4A). In contrast, chromatin accessibility increases at the *brat-F* promoter in pupae, perhaps related to the primary use of this promoter at the pupal stage. Importantly, in pupal but not larval neurons (Fig. 4A and Data S1),

**Fig. 2 Region 1 is a neural-specific enhancer, and Shep does not directly repress *brat* enhancer or promoter activities. A** ChIP-seq of H3K4me1, H3K27ac, and Shep along with STARR-seq summits throughout the *brat* locus, including the three 4C-seq differential regions in BG3 cells (red and green arrows). The *brat* promoters are defined as the 2 kb region upstream of the transcription start site of each isoform and are indicated by teal bars. **B** Luciferase constructs with the *brat-F* promoter and/or putative enhancer region co-transfected along with a Renilla construct with an *RpLII128* promoter, which serves as a transfection control for reporter assays. **C, D** Renilla-normalized luciferase expression driven by the *brat-F* promoter in BG3 or S2 cells. Region 1, but not regions 2 and 3, enhances luciferase expression in BG3 but not S2 cells. Experiments were performed with $n = 2$ biologically independent samples. **E** Normalized luciferase expression in control or Shep-depleted BG3 cells. Expression of region 1 enhancer and/or *brat-F* promoter constructs are unaffected by Shep depletion. A two-tailed Student's *t* test was used, and experiments were performed with $n = 3$ biologically independent samples. **F** Fold change of Renilla-normalized luciferase expression over *brat-A/E* promoter driven by region 1, *brat-A/E* promoter alone, or region 1 alone in BG3 cells. Cells were co-transfected with GFP (control) or *shep* dsRNA. Experiments were performed with $n = 3$ biologically independent samples. Two-tailed Student's *t* test with $^{**}p < 0.01$, $^{***}p < 0.001$. The exact *p* values are 8−e4 and 1.2−e3 for the control and Shep-kd tests, respectively. **G** Same assay as (**F**) for *brat-B/C* promoter. Experiments were performed with $n = 3$ biologically independent samples. Average values are reported as mean + SD for all luciferase assays. Two-tailed Student's *t* test with $^{*}p < 0.05$. The exact *p* values are 4−e2 and 2.7−e2 for the control and Shep-kd tests, respectively.

---

Shep depletion leads to elevated accessibility both in the vicinity of the *brat-F* promoter and at the region 4 enhancer. These results indicate close correspondence between Shep inhibition of chromatin accessibility and Shep inhibition of *brat* E–P looping and transcription in pupal neurons.

**Shep facilitates developmental closing of putative enhancers.** Our ATAC-seq profiles enabled us to identify developmentally programmed changes in chromatin accessibility of larval vs. pupal control neurons genome-wide, beyond the *brat* locus. The majority of ATAC-seq peaks (33,882 of 41,568) display differential accessibility between control larval and pupal neurons, resulting in 17,367 opening and 16,515 closing regions over this time window. In order to determine whether accessibility changes occur at enhancers genome-wide during neuronal remodeling, we performed CUT&Tag of H3K4me1 on FACS-sorted control larval and pupal neurons (*elav > Dcr-2, mCD8::GFP*) to identify all putative enhancers irrespective of activity. The intersection between ATAC-seq and CUT&Tag profiles indicated that 70% (24,155 out of 33,882) of temporal accessibility changes occur at H3K4me1-marked regions, suggesting that enhancers are particularly malleable during this developmental window. Furthermore, 86% (7353 out of 8528) of H3K4me1 regions change in accessibility during neuronal remodeling. Therefore, widespread temporal regulation of accessibility at enhancer regions is observed during normal neuronal remodeling.

Upon genome-wide examination of Shep-dependent differentially accessible regions, we found strong Shep-facilitated enhancer closing during neuronal remodeling. Overall, chromatin accessibility in Shep-depleted neurons shows substantially more changes in pupae (3147 differential regions) than in larvae (331), consistent with majority pupal-specific transcriptome changes upon Shep depletion[11]. Considering only Shep-dependent accessibility changes in pupal neurons, we found 1127 regions labeled by H3K4me1 that are normally kept inaccessible by Shep (Shep-inhibited). During normal neuronal development, these regions are accessible in larvae but become closed by the pupal stage (Fig. 4B, C). However, in Shep-depleted neurons, these regions gain additional accessibility in pupal compared to larval neurons, indicating greatly compromised closing of these putative enhancers. Intriguingly, these regions are most enriched for binding motifs for the zinc-finger DNA-binding proteins, Klu transcription factor, and GAGA-factor (GAF) (Fig. S4E, F), a protein that has been implicated in insulator activity[18,19]. While many enhancer regions fail to close during neuronal remodeling in the absence of Shep, we also identified 1333 Shep-promoted regions labeled by H3K4me1 that are inaccessible in larval neurons but open during normal neuronal remodeling. Upon Shep depletion, these regions can only be partially opened, suggesting that Shep moderately contributes to the accessibility of these regions (Fig. 4B–D). For these regions, we obtained enrichment for motifs for Klu and the Sp1 family of transcription factors[20] (Fig. S4F). By also performing CUT&Tag of

H3K27me3 with sorted control neurons from pupae, we observed more frequent Shep regulation of accessibility in either direction of H3K4me1-labeled enhancers (2560) than H3K27me3-labeled inactive chromatin regions (1305), despite similar genomic coverage by H3K4me1 (4.1e7 bp) and H3K27me3 (3.8e7 bp). These results indicate enhancer-biased Shep regulation of accessibility. Taken together, we conclude that Shep mainly mediates the closing of numerous enhancers genome-wide during neuronal remodeling.

Finally, we verified that Shep-dependent changes in chromatin accessibility are functionally associated with gene expression changes genome-wide. *Myc*, another key downstream target that is repressed by Shep in pupal neurons[11,21], displays temporally dynamic accessibility patterns at H3K4me1-enriched regions, some of which are also Shep-inhibited (Fig. S5A). Genome-wide, we found that Shep-inhibited accessibility changes are indeed statistically enriched for Shep-inhibited gene expression changes in pupal neurons (Fig. S5B). A statistically significant association between these two events is even more pronounced when restricted to H3K4me1-marked accessibility changes (Fig. 4E), suggesting that genome-wide enhancer closure is a key mechanistic step in transcriptional inhibition mediated by Shep. Among 82 genes inhibited by Shep for both expression and enhancer accessibility (Fig. 4E), Shep association is significantly enriched at enhancers of 26 genes (Fisher's exact test (FET), $p = 1.7e−8$, odds ratio = 4.5), suggesting direct Shep inhibition of both enhancer accessibility and transcription. In contrast, the correlation between Shep-promoted gene expression and Shep-regulated accessibility is either weaker (Fig. S5C, D) or absent (Fig. 4E; Fig. S5B) when enhancers are considered. These findings suggest that Shep inhibits chromatin accessibility, particularly enhancer accessibility, to temporally regulate gene expression during neuronal remodeling.

## Discussion

Shep attenuates transcription of a critical developmental regulator during a specific window of neuronal remodeling by inhibiting E–P looping, which is itself temporally controlled. Genome-wide, Shep reduces transcription of many downstream targets and limits chromatin accessibility of corresponding enhancer regions during neuronal remodeling. Although several transcription factors have been shown to inhibit E–P loop formation[6–8], loss of looping in these cases is likely an indirect result of repressed enhancer and/or promoter activities. In these studies and our own work, it remains a challenge to distinguish between cause and consequence; however, we demonstrated that Shep does not alter the ability of *brat* enhancers or promoters to drive a reporter either ex vivo or in vivo. Thus, we propose that Shep is a dedicated anti-looping factor, which functions primarily by inhibiting E–P looping to regulate *brat* expression.

We speculate that Shep antagonism of E–P looping involves regulation of other DNA-binding proteins with respect to their chromatin association, as evidenced by changes of chromatin

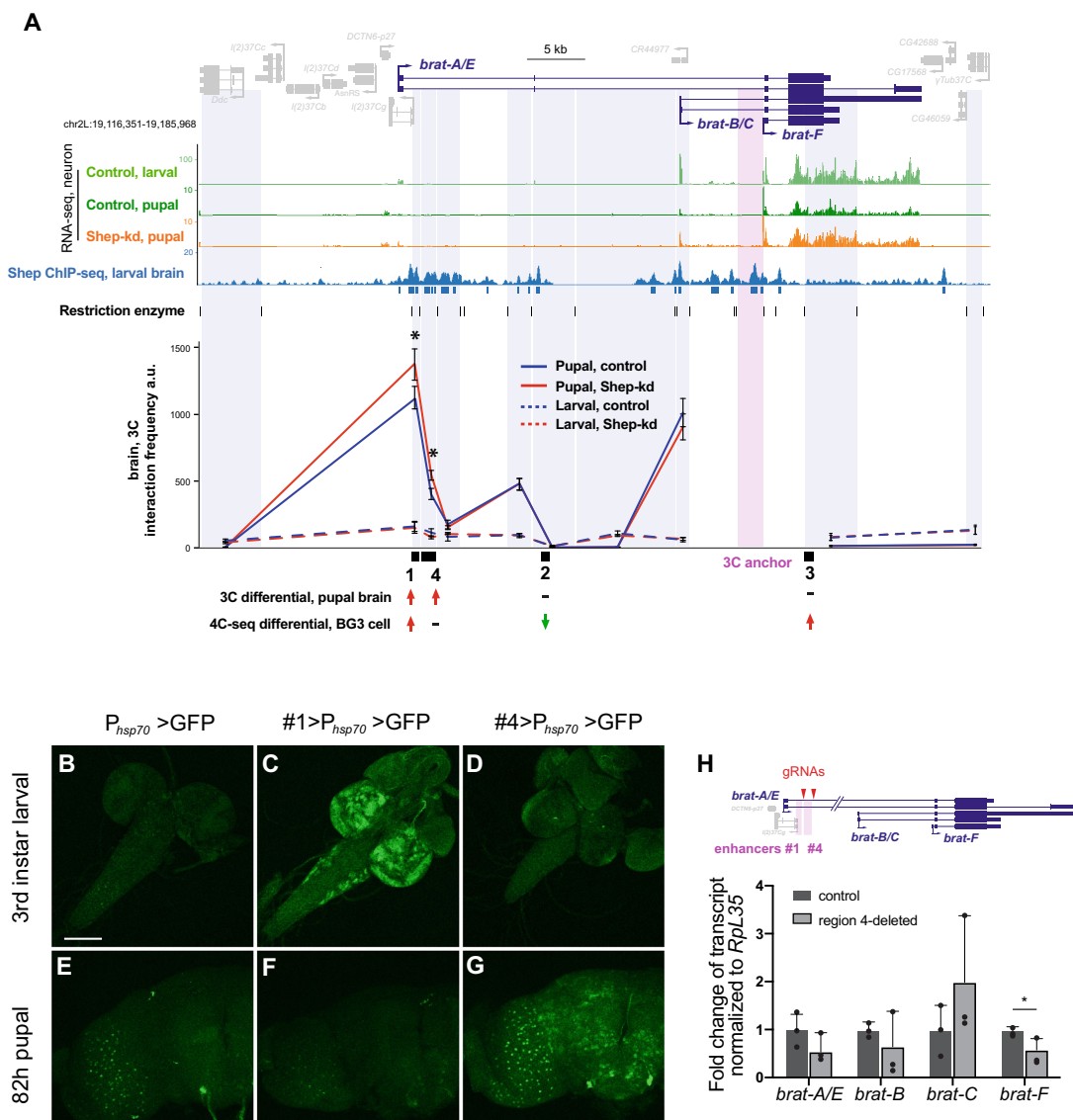

**Fig. 3 Shep inhibits *brat* E–P looping in a stage-specific manner. A** 3C interaction frequencies in control (blue) or Shep-depleted (red), larval (dashed), or pupal (solid) brains using the *brat-F* promoter as an anchor (pink shading). Digested genomic fragments for 3C quantification are labeled in purple shading. Normalized interaction frequencies in arbitrary units were analyzed by a two-tailed Student's *t* test. Statistical significance is presented as *$p < 0.05$ for three biological replicates. Data are presented as mean values ± SEM. Exact *p* values are reported in the Source Data. **B–G** Regions 1 and 4 harbor temporally regulated enhancer activities. GFP reporter expression driven by the minimal *hsp70* promoter with or without candidate enhancer regions was visualized by immunostaining larval and pupal brains with anti-GFP. Note leaky GFP expression driven by *hsp70* promoter in pupal brains (**E**). All flies were grown at 25 °C. The GFP phenotypes are reproducible across three independent repeated experiments. Scale bars: 100 μm. **H** Quantification of steady-state *brat* expression in pupal brains of homozygous CRISPR-deletion of region #4 mutants by RT-qPCR. The gRNAs used to target region 4 are labeled with arrowheads. Experiments were performed with $n = 3$ biologically independent samples. Data are presented as mean values + SEM, and a two-tailed Student *t* test was performed with *$p < 0.05$. The exact *p* value is 4.3−e2.

accessibility at enhancers. Our motif analysis of Shep-regulated enhancers identified Klu and the Sp1 family of transcription factors, two classes of zinc-finger proteins that are both enriched in the nervous system during metamorphosis[14] when neuronal remodeling occurs. We also identified enrichment of a GA-rich motif known to be bound by the insulator-associated factor GAF, and this motif is present in the region 4 enhancer. Although GAF is an attractive candidate considering that it binds the *brat-F* promoter, as well as enhancer regions 1 and 4 in chromatin of BG3 cells (Fig. S4G), the GA-rich motifs identified in our analyses are fairly generic repeated sequences that could alternatively be bound by other factors, such as CLAMP[22] or AGO2[23], for

example. Future studies are required to interrogate the potential roles of these proteins in Shep mechanistic function. In addition, Shep is known to bind transcripts of its chromatin target genes[11,24], raising the possibility that Shep may load onto chromatin during transcription and concomitantly regulate chromatin accessibility and E–P looping. Finally, specific coding or non-coding RNAs could be required for Shep function in this context, as RNA-binding is required for Shep antagonism of *gypsy* insulator activity[9]. Our results do not rule out the possibility that Shep also downregulates *brat* or other genes by posttranscriptional mechanisms. Given widespread evidence that E–P looping and enhancer accessibility are both frequently associated with gene

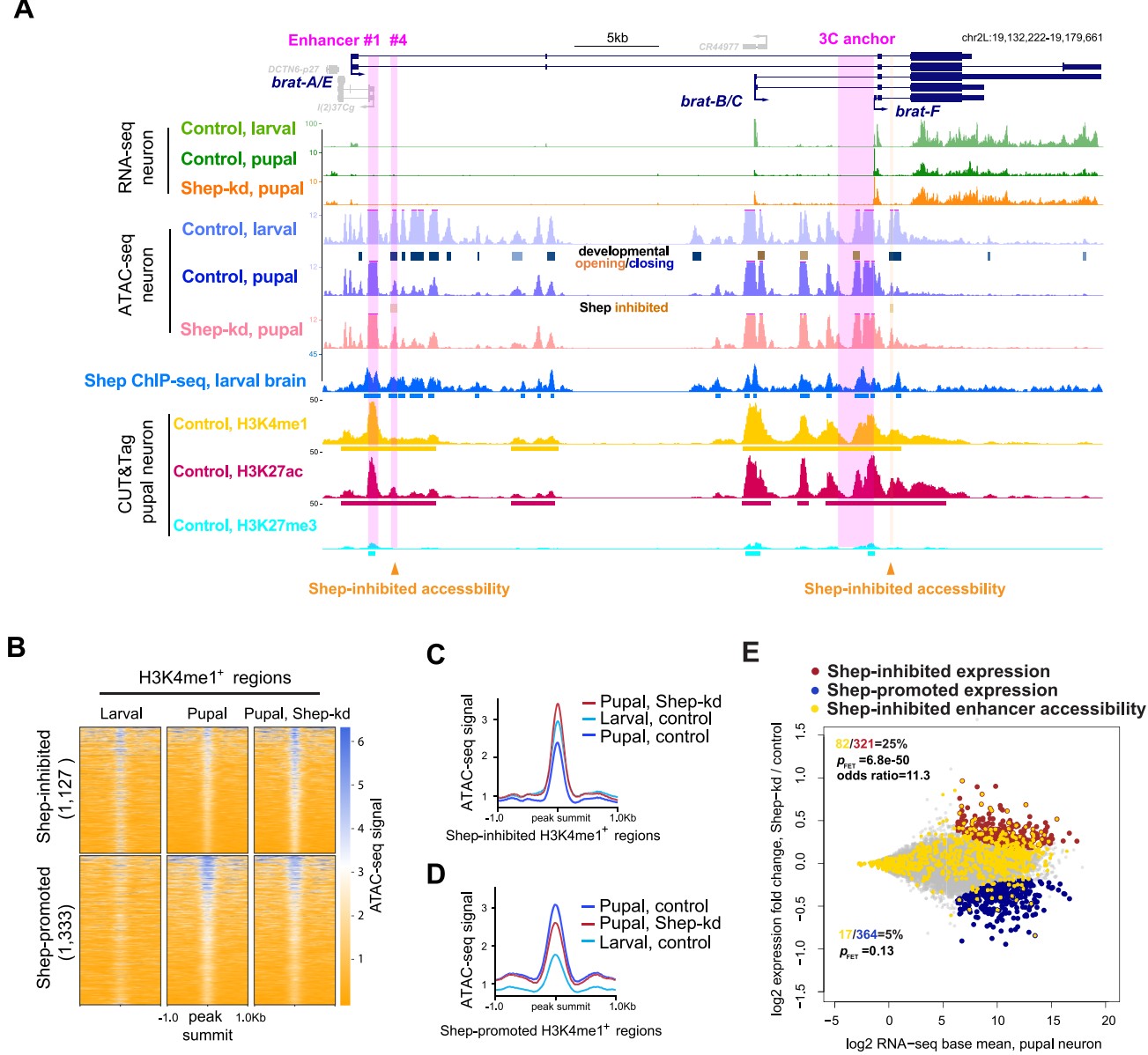

**Fig. 4 Shep mainly inhibits enhancer accessibility in pupal neurons genome-wide. A** ATAC-seq profiles of control larval, control pupal, and Shep-depleted pupal neurons at the *brat* locus. Called differentially accessible regions that open (orange) or close (blue) between control larvae and pupae are indicated. Only two accessible regions change (open) after Shep depletion in pupal neurons (orange arrowheads and shading; FDR = 0.07 and fold change = 1.3 for both regions). CUT&Tag signals of H3K4me1 (yellow), H3K27ac (red), and H3K27me3 (cyan) in sorted control pupal neurons are shown. **B** Heatmaps of ATAC-seq signals are shown for Shep-regulated ATAC-seq peak regions (one per row, sorted by overall signal) in control vs. Shep-depleted pupal neurons that are also marked by CUT&Tag signal for H3K4me1 in sorted control pupal neurons. The corresponding signal in control larval neurons is also shown. **C, D** Average ATAC-seq signals across regions in B are shown. **E** Shep inhibits enhancer accessibility to reduce gene expression in sorted pupal neurons. Genes corresponding to Shep-regulated enhancer accessibility are labeled in gold and overlaid with genes for which expression is Shep-inhibited (red) or Shep-promoted (blue). Genes unaffected by depletion of Shep are shown in grey. The numbers of genes in each group are indicated and colored accordingly, and *p* values of two-tailed FET without adjustment are reported for indicated colored groups.

activation, we predict that regulation of chromatin looping and enhancer accessibility are commonly utilized cellular mechanisms to control developmental gene expression programs.

Notably, our results provide evidence of temporal regulation of 3D chromatin organization that facilitates post-mitotic neuronal remodeling. Previous studies in *Drosophila* have identified a variety of genetic and cellular processes underlying neuronal remodeling[12,21,25], and recent studies have begun to elucidate transcriptome dynamics throughout this developmental program[11,26]. However, fundamental mechanisms regulating 3D chromatin structure that result in temporal changes in gene expression during this essential process in any

model system remain undefined. Our data reveal widespread changes in enhancer accessibility during normal neuronal maturation, similar to what has been observed during developmental differentiation in the mouse cerebellum[27], suggesting conserved regulation of enhancer accessibility during neuronal maturation. We found that Shep inhibits enhancer accessibility during the temporal progression of neuronal remodeling, which corresponds to Shep chromatin association, Shep repression of chromatin looping, and temporal inhibition of the key target *brat* gene. Moderate expression changes of all *brat* isoforms, as well as *brat* enhancer accessibility, may result from changes occurring only in specific neurons within the total isolated population

and/or transient changes in chromatin accessibility. Since we and others have observed that E–P looping is not necessarily sufficient to activate transcription[28], it is likely that additional chromatin-related events must occur in order to achieve transcriptional activation. Widespread and conserved dynamics of enhancer accessibility is observed in the nervous system across organisms[27,29,30]; likewise, regulated E–P looping may also be utilized genome-wide during neuronal remodeling to regulate temporal gene expression. Finally, our findings suggest potential mechanisms underlying functions of human Shep orthologs, of which mutation and misexpression are associated with various neurological diseases, including ALS, Alzheimer's, Parkinson's, and schizophrenia[31].

## Methods

**Fly strains**. Fly stocks and crosses were grown on standard cornmeal-yeast-agarose medium at 25 °C. Only female third instar larvae or 82 h pupae were used for experiments. Fly strains used include *elav-Gal4* (FBst0000458), *UAS-Dcr-2* (FBst0024650), and *UAS-shep-RNAi* (FBst0462204). For a generation of the *phsp70-GFP* strains, region 1 or 4 was cloned into NotI restriction sites in the pEGFP-attB plasmid (*Drosophila* Genomics Resource Center). The resulting plasmids were integrated by phiC31 integrase at the attP40 locus by BestGene. To generate flies deleted for region 4 using CRISPR/Cas9 editing, two gRNAs CACTGTGCCAGAAAGTTCCA and AATGCACTGATTAACAGTAA were inserted into the BbsI sites of the plasmid pUC57 (a gift from B. Oliver, NIDDK) for gRNA expression after embryo injection. To delete endogenous region 4, genomic sequences flanking region 4 were inserted on either side of the dsRed coding sequence into the plasmid pUC19 (a gift from B. Oliver, NIDDK) as a homologous repair template. Both plasmids were constructed by GenScript and were injected into *yw;;nos-Cas9(III-attP2)* flies by BestGene. Full plasmid sequences and primers used to validate the insertion location by sequencing are included in Data S1.

**Cell culture and transfection**. BG3-c2 cells and S2 cells (*Drosophila* Genomics Resource Center) were cultured at 25 °C in Schneider's medium with 10% FBS or M3 + BPYE medium with 10% FBS, respectively. BG3 medium was further supplemented with 10 μg/mL insulin. The MEGAscript T7 Kit (Thermo Fisher Scientific) was used to generate dsRNA, which was purified using NucAway Spin Columns (Thermo Fisher Scientific). The Amaxa Cell Line Nucleofector Kit V (Lonza) was used according to the manufacturer's protocol to transfect constructs and dsRNA into 5–10 million BG3 or S2 cells using programs T30 or G30, respectively. Two micrograms of dsRNA targeting Shep or GFP was used to deplete Shep or serves as a control. One microgram of each luciferase construct and 1 μg of Renilla control construct were co-transfected into cells to perform luciferase assays. Experiments were performed 4 d after transfection.

**Nascent RNA quantification**. Nascent RNAs were labeled and captured with a Click-iT Nascent RNA Capture Kit (Thermo Fisher Scientific C10365). Four days after transfection with dsRNA, BG3 cells were incubated with ethylene uridine for 1 h, and RNA was Trizol-extracted (ThermoFisher 15596026), biotinylated, and captured with 12 μL T1 beads according to manufacturer's protocol. RNA captured on beads was resuspended with a 20 μL reaction system containing SuperScript IV Reverse Transcriptase (Thermo Fisher Scientific 18090010) for cDNA synthesis. The cDNA was further quantified by qPCR, and primer sequences are included in Data S1.

**Cloning of luciferase constructs**. Luciferase cDNA was inserted between the XhoI and SpeI restriction sites of the pJET1.2 cloning vector. The 2 kb region upstream of *brat-A/E*, *brat-B/C*, or *brat-F* was amplified from Oregon-R genomic DNA and then inserted between the XhoI and EcoRI restriction sites upstream of the luciferase-encoding sequences. Candidate enhancer regions based on 4C-seq analysis were individually cloned and inserted upstream of the *brat-A/E*, *bratB/C*, or *brat-F* promoter at NotI restriction sites. Cloning primer sequences are listed in Data S1.

**Luciferase assays**. One and a half millilitre of cells from each transfection were spun at 600 × *g* for 10 min. Cell pellets were frozen at −80 °C until analysis. Cells were resuspended in 250 μL nuclease-free water, and 75 μL were plated in triplicate in opaque 96-well plates. Then 75 μL of Dual-Glo Reagent (Promega) was added to each well, and the plate was incubated for 10 min at RT. Firefly luminescence was measured using a Spectramax II Gemini EM plate reader (Molecular Devices). Next, 75 μL of Dual-Glo Stop & Glo Reagent (Promega) was added to each well and incubated for 10 min at RT before measuring Renilla luminescence. Two to three biological replicates were performed per experiment.

**Chromatin conformation capture (3C)**. For each replicate, 20 brains were dissected in Schneider's medium containing 10% FCS and 10 μg/mL insulin. Formaldehyde was added to a final concentration of 2%, and brains were fixed for 15 min at RT, then quenched with 0.125 M glycine for 10 min, followed by two 10 min rinses with washing

buffer (50 mM Tris, 10 mM EDTA, 0.5 mM EGTA, 0.25% Triton-X100). Fixed brains were stored in storage buffer (10 mM Tris-HCl pH = 8, 1 mM EDTA, 0.5 mM EGTA) at −80 °C until all samples were pestle-homogenized in lysis buffer [10 mM NaCl, 0.2% NP-40, 10 mM Tris pH 8, and Mini Complete tablet (Roche)] and incubated at 37 °C for 20 min. Nuclei were pelleted at 6000 × *g* for 5 min, and the incubation was repeated once more. Nuclei were washed with digestion buffer (ThermoFisher Scientific ER0932 plus 0.2% NP-40), pelleted at 6000 × *g* for 5 min, and incubated with the digestion buffer containing 0.2% NP-40, 0.1% SDS at 65 °C for 30 min. Triton X-100 was added to a final concentration of 1%, and samples were further incubated at 37 °C for 15 min. Ten percent of each sample was saved as an undigested control. Next, 200 U of BsRGI (ThermoFisher Scientific ER0932) was added to a final volume of 100 μL, and the digestion was incubated for 2 d at 37 °C. The restriction enzyme was inactivated at 65 °C for 20 min, and another 10% of the sample was saved as a digested control. The digested sample was diluted with 400 μL T4 ligation buffer (NEB M0202, 1% Triton X-100) and incubated at 37 °C for 30 min, followed by overnight incubation at 16 °C with 3 μL T4 DNA ligase (NEB M0202). Samples were purified with phenol–chloroform and used as 3C templates for Taqman-qPCR. The BAC CH321-86O1 (Chori) was digested and religated and used to generate 3C templates to normalize for primer efficiency, and primers targeting the *drl* locus were used to equalize loading across samples. A two-tailed Student's *t* test was performed at *$p < 0.05$. Sequences of the MGB (minor groove binder) Taqman probe, 3C primers, and loading control primers are included in Data S1.

**4C-seq libraries**. Generation of 4C-seq libraries in BG3 cells was performed using the brain 3C protocol except that BG3 cells were fixed with 1% formaldehyde for 10 min at RT, and the primary digestion enzyme used was Csp6I (ThermoFisher Scientific ER0211). Next, 3C templates were further processed according to a published 4C protocol[32] with DpnII (NEB R0543) as the secondary digestion enzyme. Libraries were sequenced at the NIDDK Genomics Core Facility on an Illumina HiSeq 2500. Primers used to amplify 4C-seq libraries are documented in Data S1.

**4C-seq analyses**. Reads from the 4C assay were aligned to the dm6 genome using bowtie2 v2.3.5 with default parameters. Subsequent sam files were sorted and converted into bam files using the Samtools v1.9 view and sort commands, respectively. Sorted bam files were then used to perform the 4C-seq analysis with the FourCSeq v1.18.0 software package. Next, 4C-seq peak identification was performed by creating two data frames containing the following metadata information: restriction enzymes, sequencing primers, reference genome ID, replicate information, viewpoint location, and sorted bam file names. This metadata was used to create the in silico digested reference genome, extract the location of the viewpoint, and map its reads to both the reference genome and the fragment reference. Mapped reads were then counted using the 'countFragmentOverlaps' command in a strand-specific manner. Because the restriction enzymes had already been trimmed, the *trim* parameter was set to 0, and the minimum mapping quality was set to 20. Counts from both left and right fragment ends were combined using the 'combineFragEnds' command. Spikes and PCR artifacts were removed using 'smoothCounts', and z-scores were calculated for potential peaks along the default distance from the viewpoint using 'getZScores'. Lastly, differential interacting fragments were identified using the 'addPeaks' function if at least one replicate had an adjusted *p* value of 0.01 and both replicates had a *z*-score larger than 3.

**FACS and ATAC-seq libraries**. A published FACS procedure was used to dissociate and select GFP-positive cells[11]. Cells were sorted on a FACSAria II machine at the Flow Cytometry Core of the National Heart, Lung, and Blood Institute. Omni-ATAC-seq libraries were generated according to a detailed protocol[17] with minor adjustments. Specifically, $2 \times 10^5$ sorted GFP-positive neurons were used for each of three biological replicates, and DNA was phenol–chloroform extracted. Eleven total PCR cycles were performed to amplify libraries, which were subsequently double size-selected with AMPure XP beads (first with 0.6× volume, then with 1.2× volume) (Beckman Coulter). Finally, 50 bp paired-end sequencing was performed at the NIDDK Genomics Core Facility on an Illumina NextSeq 550 with the High Output mode.

**ATAC-seq computational analyses**. Adapter sequences were trimmed from reads with cutadapt (v2.3; -a CTGTCTCTTATACACATCTCCGAGCCCACGAGAC -A CTGTCTCTTATACACATCTGACGCTGCCGACGA–minimum-length 18) and aligned to Flybase release dm6 reference with bowtie2 (v2.3.5;–very-sensitive, paired-end mode)[33]. Reads were depleted for those aligned to mitochondria (egrep -v chrM) and for multi-mapped reads with samtools (v1.9)[34]. Uniquely mapped reads were further depleted for PCR duplicates with picard (MarkDuplicates REMOVE_DUPLICATES = true) and computationally size-selected for inserts <150 bp to exclude nucleosome-related reads. ATAC-seq peak calling was performed with MACS2 (v2.2.6; pair-end mode -f BAMPE)[35], and differential accessibility was called with the R package DiffBind v2.6.6 (edgeR, FDR < 0.1 and lfc > 0.3). Motif enrichment analyses were performed with AME 5.3.3[36] and STREME 5.3.3[37], and only motifs identified by both algorithms with known binding factors were reported.

**CUT&Tag libraries and analyses**. After FACS sorting, collected neurons were used directly to generate CUT&Tag libraries following a protocol established by the Henikoff group (https://doi.org/10.17504/protocols.io.bcuhiwt6) with minor

changes. Specifically, $7.5 \times 10^4$ sorted GFP-positive neurons were used for each of the three biological replicates without crosslinking. Tn5 was purchased from Epicypher (15-1017) and used at 1:20. The primary antibodies H3K27me3 (Cell Signaling Technology 9733), H3K4me1 (Abcam ab8895), and H3K27ac (Abcam ab4729) were used at 1:50. Sixteen total PCR cycles were performed to amplify libraries, which were re-purified (using 1.1× volume of AMPure XP beads) after pooling to remove primer dimers. Then 50 bp paired-end sequencing was completed at the NIDDK Genomics Core Facility on a NextSeq 550 using high-output. Computational analyses of CUT&Tag data were performed with the same pipeline used for ATAC-seq except that there was no computational size-selection of sequencing reads, and peaks were called by MACS2 (v2.2.6) in broad mode.

**Immunostaining and imaging**. Samples were fixed with 4% paraformaldehyde for 1 h, washed 3 × 20 min with PBST (phosphate-buffered saline with 0.1% tween20), and blocked with 5% NGS for 1 h. Samples were then incubated with primary antibodies overnight at 4 °C, washed 3 × 20 min with PBST, and incubated with secondary antibodies for 1 h at RT. After 3 × 5 min washes with PBST, samples were mounted and images were taken as maximum-intensity z-series projections with a Zeiss780 confocal microscope. The GFP primary antibody (ThermoFisher A10262) was used at 1:8000. The goat anti-chicken secondary antibody (ThermoFisher A-11039) was used at 1:1000.

**Statistics**. All experiments were performed using three biological replicates unless noted. The $p$ values of Fisher's exact tests were calculated with R v3.6.1 and reported as two-tailed values with [***]$p < 0.001$, [**]$p < 0.01$, and [*]$p < 0.05$. Averaged values are reported as mean ± SEM unless otherwise noted.

**Reporting summary**. Further information on research design is available in the Nature Research Reporting Summary linked to this article.

## Data availability

The data that support this study are available from the corresponding author upon reasonable request. The 4C-seq, ATAC-seq, CUT&Tag, and ChIP-seq data have been deposited in the Gene Expression Omnibus database under accession number GSE154645. Publicly available data used in this study include H3K4me1 ChIP-chip (modENCODE accession number 2653), H3K27ac ChIP-chip (modENCODE accession number 295), GAF ChIP-chip (modENCODE accession number 2651), and STARR-seq (GEO accession number GSE49809). Source data are provided with this paper.

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

## Acknowledgements

We thank S. Wang for assistance with custom MA plots; L. Benner for guidance on CRISPR; and A. Dean, G. Blobel, and members of the Lei laboratory for comments on the paper. This work was funded by the Intramural Program of the National Institute of Diabetes and Digestive and Kidney Diseases, National Institutes of Health (DK015602 to E.P.L.), and the Pathway to Independence Award (K99HD097308-01A1 to D.C.).

## Author contributions

Conceptualization: D.C., C.E.M., and E.P.L.; Data Curation: D.C., C.E.M., and B.R.; Formal analysis: D.C., C.E.M., and B.R.; Funding acquisition: E.P.L. and D.C.; Investigation: D.C. and C.E.M.; Methodology: D.C., C.E.M., B.R., and L.H.M.; Project administration: D.C. and E.P.L.; Resources: E.P.L.; Software: D.C. and B.R.; Supervision: E.P.L.; Validation: D.C., C.E.M., and E.P.L.; Visualization: D.C., C.E.M., and B.R.; Writing—original draft: D.C.; Writing—review and editing: D.C., C.E.M., B.R., and E.P.L.

## Competing interests
The authors declare no competing interests.
