## [Peer Review File · Nature Communications]

REVIEWER COMMENTS

Reviewer #1 (Remarks to the Author):

Ms. Ref. No.: NCOMMS-20-35501

Title: Temporal regulation of neuronal remodeling by a chromatin anti-looping factor

Authors: Dahong Chen, Catherine E. McManus, Behram Radmanesh and Elissa P. Lei

Overview and general recommendation:

This work builds upon the authors' previous reports that:

1) the *Drosophila* RNA-binding protein Shep directly antagonizes gypsy insulator activity, leading the authors to speculate that Shep could impact 3D genome folding (Matzat et al., 2012);
2) shep-RNAi resulted in small (<2-fold) changes in mRNA levels of several hundred genes in pupal neurons but less than a hundred genes in larval neurons (Chen et al., 2017). *brat* (a cell fate determinant during neuroblast division) was weakly upregulated in shep-RNAi pupal, but not larval, neurons and is the major focus of the present manuscript.

The aim of the present study is therefore to test whether Shep regulates *brat* expression in pupal neurons by antagonizing looping of *brat* to its enhancers (pg.2 lines 20-22: "We therefore speculated that Shep may antagonize *brat* E-P looping in order to repress its transcription during metamorphosis").

To test this hypothesis, 4C-seq with a viewpoint in *brat* was performed in nervous system-derived cultured cells (BG3), in control versus shep-RNAi conditions (in which *brat* mRNA levels are increased by 1.2-fold). 3 regions had differential interaction frequencies with *brat* upon shep-RNAi (Fig. 1). These regions were tested in luciferase reporter assays in S2 (mesodermal origin) and BG3 (nervous system origin) cells. A region contacted by *brat* in BG3 cells less frequently upon shep-RNAi was found to be a BG3-specific enhancer, and its activity was not affected by shep-RNAi (Figs. 2 and S2). This region may be a distal *brat* enhancer, though this was not confirmed. *brat* chromatin contacts were assessed in vivo by 3C-qPCR in control and shep-RNAi larval or pupal brains (Fig. 3). The aforementioned putative *brat* distal enhancer (Figs. 1 and 2) interacted with *brat* only in pupal (not larval) brains, and this interaction was reduced upon shep-RNAi (Fig. 3). Does Shep regulate enhancer activity in vivo? ATAC-seq was performed on FACS-isolated neurons from control and shep-RNAi larvae or pupae. The putative *brat* distal enhancer was not differentially accessible in any sample, but the authors argue that the enhancer was close (within 1 kb) to a differentially accessible ATAC-seq peak of unknown function (Fig. 4); the relevance of this observation was not explored further. Nevertheless, the authors conclude that Shep is a "novel, dedicated chromatin anti-looping factor". Looking genome-wide beyond *brat*, shep-RNAi led to a few hundred differentially accessible regions in larvae and an order of magnitude more changes in pupae (Fig. 4). Regions that were less "closed" in shep-RNAi versus control pupal neurons often (~30%) overlapped an H3Kme1 ChIP-seq peak in BG3 cells, suggesting that they were enhancers. These putative enhancers were often more "open" in larval versus pupal control neurons (Fig. 4). This may intriguingly suggest that Shep regulates chromatin accessibility of larval enhancers in the larval-to-pupal transition. Inefficient closing of these enhancers in shep-RNAi pupal neurons is somewhat correlated with failure to downregulate gene expression during the larval-to-pupal transition (Fig. S3).

On the one hand, I found the paper to be well written and easy to follow. I find the ATAC-seq in isolated neurons of larval and pupal brains an elegant dataset and I was intrigued by the chromatin accessibility dynamics between developmental time points and by the preliminary results suggesting that Shep may regulate chromatin accessibility in the larval-to-pupal transition. On the other hand, I found that this dataset was not sufficiently explored. Most importantly, however, I find that the authors' main conclusion that Shep represses *brat* transcription by preventing enhancer-promoter looping is not supported by the current data. Therefore, I recommend that a major revision is warranted because I do not feel that this work sufficiently advances understanding in the field as it stands. I explain my concerns in more detail below. I ask that the authors specifically address each of

my comments in their response.

Major comments:

1. My main concern is that the authors' general conclusion (pg. 7 line 7: "Our results report a novel, dedicated chromatin anti-looping factor) and manuscript title (pg. 1 line 1: "Temporal regulation of neuronal remodeling by a chromatin anti-looping factor") are not supported by the data. Shep did not affect the activity nor the chromatin accessibility of a putative brat enhancer identified in this study, but the altered interaction frequencies measured by 4C-seq or 3C-qPCR upon shep-RNAi in BG3 cells or in pupal neurons could simply be a consequence of weakly altered brat transcription. In my view this study neither addresses whether Shep is directly involved in chromosomal loop formation, nor whether increased contact frequency between brat promoter and its putative distal enhancer is relevant for brat transcript levels.

It is also unclear to me why the authors generally hypothesize that Shep antagonizes larval enhancers during the larval-to-pupal transition by repressing enhancer-promoter loops (pg. 8 lines 5-7): "We found that Shep inhibits enhancer accessibility during the temporal progression of neuronal maturation, which corresponds to Shep repression of chromatin looping and expression of downstream transcriptional targets." There is no data in the manuscript that would support a global role of Shep in antagonizing enhancer-promoter looping.

2. The authors hypothesize that a differentially accessible ATAC-seq peak close (within 1 kb) to the brat putative distal enhancer may be implicated in differential contacts between brat and the enhancer. [pg. 5 lines 15-: "In pupal but not larval neurons, Shep depletion led to elevated accessibility of two specific regions, adjacent to the brat-F promoter and region 1 enhancer, indicating correspondence between increased chromatin accessibility and increased looping frequency in Shep-depleted neurons."] The authors should test their hypothesis by deleting this ATAC-seq peak and assessing whether this affects the proximity between brat and the putative distal enhancer.

3. The authors describe matching neuronal RNA-seq (previously published in Chen et al., 2017) and ATAC-seq (this study) datasets at two developmental time points, in both control and shep-RNAi animals. It would be worthwhile to exploit these datasets more deeply to clarify whether and how Shep may regulate target genes during the larval-to-pupal transition. I am not convinced that it is valid to classify ATAC-seq peaks in larval and pupal neurons as putative enhancers and non-putative enhancers based on whether they overlap H3K4me1 ChIP-seq peaks in BG3 cells (a different cell type than the one in which ATAC-seq was performed), also because I feel this leads to a large underestimation of enhancers. By this definition, only ~1/3rd of all ATAC-seq peaks are enhancers (Fig. 4B) whereas I would guess that a much larger proportion of differential ATAC-seq peaks are putative enhancers.

Minor comments:

4. Pg. 7 line 19-pg. 8 line 1: "Notably, our results provide the first evidence of temporal regulation of 3D chromatin organization that facilitates neuronal maturation. (...) However, fundamental mechanisms regulating 3D chromatin structure resulting in temporal changes in gene expression during this essential process in any model system remain undefined." Don't among several publications, that of Bonev et al., 2017 describing transcription and 3D chromatin organization changes during mouse neural differentiation in vitro and in vivo do this?

Reviewer #2 (Remarks to the Author):

In this paper, Chen D. et al characterized a role for the chromatin-associated factor Shep in regulating stage-specific enhancer-promoter interaction with focus on the *brat* gene locus. Using chromatin capture techniques (4C-seq and 3C), the authors identified an enhancer of *brat* that interacts with the *brat* promoter and showed that shep knockdown in the pupal brain could increase the enhancer-promoter interaction. Further the authors performed ATAC-seq from control and shep-KD pupal neurons and found changes in chromatin accessibility in enhancers marked by H3K4me1. Based on these evidences, the authors concluded that Shep is a tissue-specific, stage-specific anti-looping factor and that its activity is required for neuronal remodeling.

The manuscript is clearly written, and the results are well organized. The observation that knockdown of shep could increase the interaction frequency between the *brat* promoter and enhancer is interesting. However, the evidences are still weak to support key conclusions, and more mechanistic studies are needed to demonstrate causative links between Shep binding, enhancer-promoter interaction and chromatin status change. My specific points are listed below.

Major points:

1. In Fig 1A, the RNA-seq tracks cannot show a clear change in expression of different *brat* transcripts. Experiments like isoform-specific RT-qPCR will be more quantitative.
2. The signal specificity of Shep ChIP-seq in the *brat* locus should be validated by ChIP-qPCR in control and shep-kd cells.
3. In the lower panel of fig1A, it is not clear how the authors determine region 1, 2 and 3 that had significant change. Just by looking at the tracks, many other regions also changed signal. A more quantitative way with statistics to show the 4c-seq differential between control and shep KD is needed.
4. Data presented in fig2 simply indicates that the region 1 fragment works as an enhancer in BG3 cells. This is a typical enhancer reporter assay, and the enhancer activity is not dependent on Shep. I do not see clear evidence showing "Shep inhibits *brat* promoter looping with a neural enhancer" as stated in the title of fig2. Moreover, a reporter that contains a *brat*-unrelated but active enhancer in BG3 cells and the *brat* promoter should be used as a control to demonstrate specific interaction between the *brat* promoter and enhancer 1.
5. In fig.3, the authors showed shep-KD decreased the interaction frequency between the enhancer region 1 and the *brat*-F promoter. This does not necessarily mean that Shep inhibits enhancer-promoter looping, because similar effect could result from knocking down a typical repressor.
6. In fig.3, the Shep ChIP-seq data is from BG3 cells, a cell line that originated from fly larval CNS. If Shep directly regulates *brat* in a stage-specific manner, the binding of Shep to the *brat* locus should be stage-specific: strong in the larval brain but weak or gone in the pupal brain. The authors can examine this by using ChIP-qPCR in control larva, control pupa and shep-KD pupa.
7. In fig. 4A, the authors highlighted two regions with Shep-inhibited accessibility. These regions do not overlap with enhancer #1, the *brat*-F promoter or any Shep ChIP-seq peak. What are these regions, other enhancers? The overall ATAC-seq signal in these two regions is low in pupal neurons even after shep-KD, compared to that of the enhancer and promoter regions, and also much lower than that of the same region at larval stage. And the increase upon shep-KD seems marginal. Similar mild change is shown in the *Myc* locus (Fig. S3A). These results indicate that the effect on chromatin accessibility by shep-KD could be indirect and mild. I don't see clear evidence that supports the conclusion in Page 5 "..., indicating correspondence between increased chromatin accessibility and increased looping frequency in Shep-depleted neurons".
8. H3K27Ac gives better enhancer prediction than H3K4me1 do. Have the authors used the H3K27Ac-predicted enhancers for the correlation study in Fig. 4B-E?
9. In fig. 4B-E, it is hard to tell how many of the Shep-inhibited and promoted sites are directly regulated by Shep. Have the authors incorporate Shep ChIP-seq data in the analyses, for example, to see whether only the Shep-bound genes changed accessibility?

Minor points:

1. In fig. 1A, the 4C anchor point seems on the Shep ChIP-seq peak but not on the brat-F promoter. It is hard to tell the distance between the 4C anchor and the brat-F promoter with the presented scale.
2. In fig. 3, the 3C/4C anchor (covering almost two ChIP-seq peaks) looks much broader than the 4C anchor (covering a half peak) shown in fig. 1. Does this mean the 3C anchor is a much broader region?

Reviewer #3 (Remarks to the Author):

Previously, the Lei lab have shown that the conserved RNA-binding protein Shep inhibits brat expression during neuronal remodelling in the Drosophila pupal brain. Here, Chen and colleagues describe their novel findings, where they propose that Shep inhibits brat expression by antagonising the long-range interaction of the brat isoform F (brat-F) promoter with a newly identified enhancer.

Key points to address:

- i) The Shep- dependent looping from enhancer 1 to the Brat-F promoter data is both convincing and an exciting finding. However, it is not clear to what degree this impacts on brat-F expression, and the expression of other isoforms. brat-A/E, brat-B/C and brat-F all contain Shep ChIP-seq peaks just upstream of their promoters, raising the possibility that Shep could bind to these upstream regions to directly inhibit transcription. There is some indication that this could be happening from the RNA-seq data shown in Fig.1, as it appears that the longer isoforms (brat-E, and possibly brat-C) show increased expression upon Shep KD. If Shep is only regulating expression via inhibiting looping to the brat-F promoter, then levels of brat-E/C would be unaffected, raising the possibility that Shep is either regulating E-P looping to brat-E/C, and/or it is acting directly to repress transcription. The in vitro experiments with the brat-F promoter (which contains a Shep binding site) do at least seem to argue against direct repression by Shep, however, it is not known for the other isoforms. Testing the activity of the brat-A/E and brat-B/C promoters in a Shep-KD condition, in vitro, could help rule this out. qPCR of brat-F (using the unique first exon) might also help.
- ii) How can the authors explain why the changes in chromatin accessibility are observed at regions neighbouring the identified regulatory regions, rather than at those regions themselves (Fig 4)?

Minor points:

- i) It would be helpful if the brat-F promoter region in the in vitro experiments was clearly defined in the main text and in the corresponding Fig 2A. Annotation of the brat-F promoter region on the figure would be helpful to provide context (i.e. relative position to enhancers and whether it contains a Shep binding site).
- ii) Figures 4B and C are very confusing and not easily interpreted.
- iii) There is good evidence that brat regulation is important for neuronal remodelling, however, it is less clear for general neuronal maturation. I suggest that the text is re-worded to only mention remodelling, in order to prevent misinterpretation/confusion.

Tony Southall

Ms. Ref. No.: NCOMMS-20-35501

Title: Temporal regulation of neuronal remodeling by a chromatin anti-looping factor

Authors: Dahong Chen, Catherine E. McManus, Behram Radmanesh, Leah H. Matzat, and Elissa P. Lei

We thank the reviewers for their thoughtful comments, each of whom raised important points. We have addressed all of the concerns as detailed below. In summary, the main concerns were:

- 1) Regions that were identified as novel distal *brat* enhancers in the original study were not convincingly shown to act as enhancers *in vivo*.
- 2) Shep may simply be acting as a repressor of either enhancer or promoter activities, and antagonism of E-P looping may be a secondary effect.
- 3) Causative links between Shep binding, enhancer-promoter interaction, and chromatin status change were not sufficiently demonstrated.
- 4) Insufficient information regarding *brat* isoform promoter regulation was provided.

To address these concerns, we used a combination of *ex vivo* luciferase assays and *in vivo* enhancer trap experiments to show that these regions do indeed act as enhancers and drive broad expression in brains, consistent with known overall *brat* expression patterns. Furthermore, by depleting Shep in combination with these assays, we demonstrated that Shep does not directly regulate either promoter or enhancer activities *ex vivo* or *in vivo*. These results suggest that Shep does not simply function as a generic transcription repressor to inactivate *brat*. Moreover, we performed additional neuron- and brain-specific CUT&Tag and ChIP-seq profiling of histone modification marks and Shep, respectively. These new experiments allowed us to directly compare Shep-dependent gene expression, E-P looping, and chromatin accessibility with Shep chromatin association in appropriately matched tissues and/or stages of development. Finally, we interrogated the remaining *brat* isoform promoters with respect to gene expression and E-P looping in control and Shep-depleted flies. As a result of these revisions, our manuscript is greatly improved and strongly supports our original model that Shep acts as a dedicated antagonist of E-P looping to inhibit *brat* expression during neuronal remodeling.

REVIEWER COMMENTS

Reviewer #1:

Overview and general recommendation:

This work builds upon the authors' previous reports that:

- 1) the *Drosophila* RNA-binding protein Shep directly antagonizes gypsy insulator activity, leading the authors to speculate that Shep could impact 3D genome folding (Matzat et al., 2012);
- 2) shep-RNAi resulted in small (<2-fold) changes in mRNA levels of several hundred genes in pupal neurons but less than a hundred genes in larval neurons (Chen et al., 2017). *brat* (a cell fate determinant during neuroblast division) was weakly upregulated in shep-RNAi pupal, but not larval, neurons and is the major focus of the present manuscript.

The aim of the present study is therefore to test whether Shep regulates *brat* expression in pupal neurons by antagonizing looping of *brat* to its enhancers (pg.2 lines 20-22: “We therefore speculated that Shep may antagonize *brat* E-P looping in order to repress its transcription during metamorphosis”).

To test this hypothesis, 4C-seq with a viewpoint in *brat* was performed in nervous system-derived cultured cells (BG3), in control versus shep-RNAi conditions (in which *brat* mRNA levels are increased by 1.2-fold). 3 regions had differential interaction frequencies with *brat* upon shep-RNAi (Fig. 1). These regions were tested in luciferase reporter assays in S2 (mesodermal origin) and BG3 (nervous system origin) cells. A region contacted by *brat* in BG3 cells less frequently upon shep-RNAi was found to be a BG3-specific enhancer, and its activity was not affected by shep-RNAi (Figs. 2 and S2).

In our original manuscript, we actually showed that this enhancer region (#1) interacts **more** frequently with the *brat-F* promoter, in both BG3 cells and brains depleted for Shep (Figure 1A and E, Figure 3A).

We would like to point out that in our revised manuscript, we adjusted our quantitative Taqman 3C-qPCR assays by further normalizing interaction values to a standard curve derived from a digested and re-ligated BAC template control. This additional control allows correction for differences in PCR efficiency across different ligation templates (Figure 3A). This more rigorous 3C quantification did not change any of our original conclusions and allows for more accurate direct comparison of 3C values across the locus in our revised manuscript.

This region may be a distal *brat* enhancer, though this was not confirmed.

In order to address the reviewer’s concern, we now provide three additional main lines of evidence to support the conclusion that region #1 functions as distal *brat* enhancer *in vivo*:

In our original manuscript, we demonstrated that region #1 activates *brat-F* promoter expression in BG3 cells (Figure 2). In our revised manuscript, we additionally confirmed region #1 enhancer activity *in vivo* by cloning the element upstream of an *hsp70* promoter-driven GFP reporter (Figure 3). Resultant GFP expression is observed in the brain, overlapping the known *in vivo* expression pattern of *brat* (Chen et al., *Development*, 2018).

Furthermore, we already showed in BG3 cells using existing Modencode datasets that region #1 is coated with histone modification marks (H3K4me1 and H3K27ac), which correlate with enhancer presence and activity, respectively. The revised version of the manuscript adds CUT&Tag data for both histone marks, which shows that region #1 is also similarly coated in pupal neurons (Figure 4A). We also performed CUT&Tag for the repressive histone mark H3K27me3, which is low throughout the *brat* locus but is mildly present specifically at region #1. Interestingly, region 1 is only active in larvae (Figure 3C) but not pupae (Figure 3F) despite looping to the *brat-F* promoter (Figure 3A). Note that E-P looping is known not to be sufficient for transcriptional activation (Ghavi-Helm et al., *Nature Genetics*, 2019)

To further confirm distal enhancer activity of region #1, we transfected nuclease-deactivated Cas9 fused to the transcriptional activator VP64 (Lin et al., *Genetics*, 2015) along with RNA guides specific to region #1 (see embedded Figure Aa) in order to transcriptionally activate this region using at least three biological replicates. RT-qPCR performed on total RNA showed significantly increased expression of *brat-B* and *brat-C*, which are already highly expressed in BG3 cells, as a result of Cas9-VP64 mediated activation. We also observed increased *brat-F* expression that is not statistically significant, presumably due to very low and relatively variable expression of *brat-F* in control transfected BG3 cells (see embedded Figure Ab). Isoforms *brat-A* and *brat-E* were not interrogated due to the proximity of their promoter to the targeted region. Therefore, ectopic activation of region #1 by dCas9-VP64 increases expression of each of the distal *brat* promoters despite at least 23 kb distance. Since our manuscript focuses on the *brat-F* promoter, we chose not to add these data to the manuscript. It is important to note that transfected BG3 cells display aberrant growth after being electroporated with the large amounts of gRNA and dCas9-VP64 expression plasmids that are required to activate *brat*.

Figure A. Activation of region 1 enhancer increases *brat* expression. a) gRNA target is labeled by the arrow head at region 1 b) qPCR quantification of *brat* isoforms after activating region 1

brat chromatin contacts were assessed in vivo by 3C-qPCR in control and shep-RNAi larval or pupal brains (Fig. 3). The aforementioned putative *brat* distal enhancer (Figs. 1 and 2) interacted with *brat* only in pupal (not larval) brains, and this interaction was reduced upon shep-RNAi (Fig. 3). Does Shep regulate enhancer activity in vivo?

This enhancer region (#1) actually interacts **more** frequently by 3C with the *brat-F* promoter upon Shep depletion (Figure 3A). Our new data show that a neighboring region (#4) to region #1 also interacts with the *brat-F* promoter more frequently in pupal brains upon Shep depletion. Both regions #1 and #4 display enhancer activity *in vivo* when trapped by an *hsp70* promoter-driven GFP reporter (Figure 3C and G). We show that Shep does not directly regulate activities of either #1 or #4 enhancers *in vivo*, since strong loss-of-function mutation of Shep (*shep*^{BG00836}) does not affect enhancer activities in this assay (Figure S4A-D).

ATAC-seq was performed on FACS-isolated neurons from control and shep-RNAi larvae or pupae. The putative *brat* distal enhancer was not differentially accessible in any sample, but the authors argue that the enhancer was close (within 1 kb) to a differentially accessible ATAC-seq peak of unknown function (Fig. 4); the relevance of this observation was not explored further.

Our new data suggest that region #1 is an enhancer in BG3 cells and larval brains, whereas we show using a GFP enhancer trap assay that the nearby Shep-dependent differentially accessible region #4 is an enhancer that is active specifically in pupal brains (Figure 3B-G). We also extended our 3C analysis in brains and found that looping between region #4 and the *brat-F* promoter is inhibited by Shep specifically in pupal brains (Figure 3A). These new data

suggest that Shep temporally inhibits *brat* enhancer accessibility and E-P looping to repress *brat* expression in pupal brains.

Nevertheless, the authors conclude that Shep is a “novel, dedicated chromatin anti-looping factor”. Looking genome-wide beyond *brat*, shep-RNAi led to a few hundred differentially accessible regions in larvae and an order of magnitude more changes in pupae (Fig. 4). Regions that were less “closed” in shep-RNAi versus control pupal neurons often (~30%) overlapped an H3Kme1 ChIP-seq peak in BG3 cells, suggesting that they were enhancers. These putative enhancers were often more “open” in larval versus pupal control neurons (Fig. 4). This may intriguingly suggest that Shep regulates chromatin accessibility of larval enhancers in the larval-to-pupal transition. Inefficient closing of these enhancers in shep-RNAi pupal neurons is somewhat correlated with failure to downregulate gene expression during the larval-to-pupal transition (Fig. S3).

On the one hand, I found the paper to be well written and easy to follow. I find the ATAC-seq in isolated neurons of larval and pupal brains an elegant dataset and I was intrigued by the chromatin accessibility dynamics between developmental time points and by the preliminary results suggesting that Shep may regulate chromatin accessibility in the larval-to-pupal transition. On the other hand, I found that this dataset was not sufficiently explored.

We appreciate the reviewer’s enthusiasm for our ATAC-seq datasets across developmental time points with and without Shep depletion and agree that this rich dataset deserved further examination. Our original analysis identified thousands of genomic loci that display temporally-regulated accessibility during neuronal remodeling. We further explored these loci by additionally performing CUT&Tag of H3K4me1 with sorted pupal neurons to locate thousands of enhancers that display dynamic accessibility during normal neuronal remodeling (Figure 4B-D).

Furthermore, in the original manuscript, we intersected our pupal neuronal ATAC-seq data with RNA-seq data and found a strong correlation between Shep-regulated gene accessibility and gene expression (Figure S5B and C). In the revised manuscript, we again intersected these Shep-regulated genes with our new H3K4me1 CUT&Tag and Shep ChIP-seq data and identified strong enrichment of Shep binding at enhancers that are regulated by Shep for both accessibility and expression of their proximal genes (Figure 4E, Figure S5D).

Finally, we performed motif enrichment analysis of these Shep-regulated enhancers using AME (McLeay et al., *BMC Bioinformatics*, 2010) and STREME (Bailey et al., *bioRxiv*, 2020) algorithms. Among the enhancers inhibited but not promoted by Shep for accessibility in pupal neurons, we identified the strongest enrichment for the binding motif of the early growth response protein Klu and an insulator-related protein GAF (Figure S4E). The possible significance of these results is discussed on pp. 10-11.

These additional analyses significantly extend our understanding of Shep function during neuronal remodeling and provide strong support for direct Shep regulation of enhancer accessibility and gene expression.

Most importantly, however, I find that the authors' main conclusion that Shep represses *brat* transcription by preventing enhancer-promoter looping is not supported by the current data.

With the new data described above, both our *ex vivo* and *in vivo* findings confirmed that Shep inhibits E-P looping and expression of *brat*. Furthermore, our reporter assays in both BG3 cells and brains in control compared to Shep depletion strongly refute the possibility that Shep simply represses either *brat* promoters or enhancers. Therefore, we conclude that Shep does not act as a general transcription repressor, but instead, acts as an anti-looper to repress *brat* expression at the pupal stage.

Therefore, I recommend that a major revision is warranted because I do not feel that this work sufficiently advances understanding in the field as it stands. I explain my concerns in more detail below. I ask that the authors specifically address each of my comments in their response.

Major comments:

1. My main concern is that the authors' general conclusion (pg. 7 line 7: "Our results report a novel, dedicated chromatin anti-looping factor) and manuscript title (pg. 1 line 1: "Temporal regulation of neuronal remodeling by a chromatin anti-looping factor") are not supported by the data. Shep did not affect the activity nor the chromatin accessibility of a putative *brat* enhancer identified in this study, but the altered interaction frequencies measured by 4C-seq or 3C-qPCR upon shep-RNAi in BG3 cells or in pupal neurons could simply be a consequence of weakly altered *brat* transcription. In my view this study neither addresses whether Shep is directly involved in chromosomal loop formation, nor whether increased contact frequency between *brat* promoter and its putative distal enhancer is relevant for *brat* transcript levels.

As stated above, we showed that Shep does not regulate *brat* promoter or enhancer activities in BG3 cells in our original manuscript. Our new data shows that the Shep-dependent differentially accessible region at *brat* (named region #4) also displays enhancer activity in pupal brains that does not depend on Shep (Figure S4). Since enhancer and promoter activities function independently of Shep, and Shep inhibits *brat* E-P looping, we suggest that Shep reduces *brat* transcription by antagonizing E-P chromatin looping.

As a further test, we CRISPR-deleted the region #4 enhancer in flies and observed specifically reduced *brat-F* expression in pupal brains (Figure 3H), indicating that this distal enhancer activates *brat-F* expression *in vivo*.

Also as mentioned above, transfection of dCas9-VP64 along with a guide RNA specific to the region #1 enhancer into BG3 cells leads to increased distal *brat-B/C* and *brat-F* isoform expression (see response to reviewer 1 above, embedded Figure A).

Furthermore, we showed that Shep associates with chromatin at the *brat-F* promoter both *ex vivo* and *in vivo* (Figures 1 and 3) as well as at the region #4 enhancer *in vivo* (Figure 3), consistent with our model that Shep directly inhibits enhancer accessibility and looping to the *brat* promoter in order to repress *brat* expression.

Taken together, our results strongly support our proposed model that Shep antagonizes E-P looping at *brat* to downregulate expression at the pupal stage of neuronal remodeling.

It is also unclear to me why the authors generally hypothesize that Shep antagonizes larval enhancers during the larval-to-pupal transition by repressing enhancer-promoter loops (pg. 8 lines 5-7): “We found that Shep inhibits enhancer accessibility during the temporal progression of neuronal maturation, which corresponds to Shep repression of chromatin looping and expression of downstream transcriptional targets.” There is no data in the manuscript that would support a global role of Shep in antagonizing enhancer-promoter looping.

We have modified the Discussion on p. 7 to clarify that Shep antagonism of E-P looping may only occur at the *brat* locus.

2. The authors hypothesize that a differentially accessible ATAC-seq peak close (within 1 kb) to the *brat* putative distal enhancer may be implicated in differential contacts between *brat* and the enhancer. [pg. 5 lines 15-: “In pupal but not larval neurons, Shep depletion led to elevated accessibility of two specific regions, adjacent to the *brat-F* promoter and region 1 enhancer, indicating correspondence between increased chromatin accessibility and increased looping frequency in Shep-depleted neurons.”] The authors should test their hypothesis by deleting this ATAC-seq peak and assessing whether this affects the proximity between *brat* and the putative distal enhancer.

Our original manuscript proposed that Shep inhibits region #1 enhancer looping to the *brat-F* promoter. Our revised manuscript shows that region #1 displays enhancer activity specifically in larval brains (Figure 3C). Furthermore, our new data show that the neighboring differentially accessible ATAC site (region #4) exhibits enhancer activity and is also inhibited by Shep with respect to looping to the *brat-F* promoter in pupal brains (Figure 3A and G).

As the reviewer suggested, we first attempted to delete region #4 by excising a P-element insertion (FlyBase strain *FBst0324214*) located at region #4. We obtained >30 independent strains harboring excision events, but all of these mutants turned out to be homozygous lethal, presumably due to imprecise excision that disrupts the proximal promoter of the essential gene *l(2)37Cg* encoded nearby on the opposite strand (see Figure 3A for location). Therefore, this excision approach failed.

However, as mentioned above, we were successful with CRISPR-deletion of the region #4 enhancer. In homozygous deletion mutants, we found that *brat-F* expression is reduced in pupal brains (Figure 3H). These results strongly support our model that Shep inhibits both E-P looping and enhancer accessibility to regulate *brat* expression.

3. The authors describe matching neuronal RNA-seq (previously published in Chen et al., 2017) and ATAC-seq (this study) datasets at two developmental time points, in both control and shep-RNAi animals. It would be worthwhile to exploit these datasets more deeply to clarify whether and how Shep may regulate target genes during the larval-to-pupal transition. I am not convinced that it is valid to classify ATAC-seq peaks in larval and pupal neurons as putative

enhancers and non-putative enhancers based on whether they overlap H3K4me1 ChIP-seq peaks in BG3 cells (a different cell type than the one in which ATAC-seq was performed), also because I feel this leads to a large underestimation of enhancers. By this definition, only ~1/3rd of all ATAC-seq peaks are enhancers (Fig. 4B) whereas I would guess that a much larger proportion of differential ATAC-seq peaks are putative enhancers.

We agree with the reviewer that it is not ideal to compare different datasets from one cell type to another. So, as stated above, we performed CUT&Tag of H3K4me1 on sorted pupal neurons to identify neuronal enhancers and updated all of our integrated analyses of differential accessibility using this new matching dataset. Use of matched material resulted in a much higher corresponding proportion of temporal accessibility changes at pupal neuronal enhancers (~70% of changed accessible sites are enhancers whereas only 44% are marked by H3K27me3) than we originally calculated using the unmatched BG3 H3K4me1 profile (~30%). We also explored our ATAC-seq data in more detail and found a strong correlation between Shep-inhibited enhancer accessibility and Shep-dependent gene expression (Figure 4E). These findings are consistent with the possibility that Shep-mediated enhancer closure during remodeling is related to the underlying mechanism behind Shep-dependent temporal regulation of gene expression.

Minor comments:

4. Pg. 7 line 19-pg. 8 line 1: “Notably, our results provide the first evidence of temporal regulation of 3D chromatin organization that facilitates neuronal maturation. (...) However, fundamental mechanisms regulating 3D chromatin structure resulting in temporal changes in gene expression during this essential process in any model system remain undefined.” Don’t among several publications, that of Bonev et al., 2017 describing transcription and 3D chromatin organization changes during mouse neural differentiation in vitro and in vivo do this?

The Bonev et al., 2017 paper studies neural differentiation, but our manuscript focuses on post-mitotic neuronal remodeling. Neuronal remodeling occurs after neuronal differentiation, so these two stages of development are distinct. We have updated the Introduction on pp. 2-3 to make this point clearer.

Reviewer #2:

In this paper, Chen D. et al characterized a role for the chromatin-associated factor Shep in regulating stage-specific enhancer-promoter interaction with focus on the brat gene locus. Using chromatin capture techniques (4C-seq and 3C), the authors identified an enhancer of brat that interacts with the brat promoter and showed that shep knockdown in the pupal brain could increase the enhancer-promoter interaction. Further the authors performed ATAC-seq from control and shep-KD pupal neurons and found changes in chromatin accessibility in enhancers marked by H3K4me1. Based on these evidences, the authors concluded that Shep is a tissue-specific, stage-specific anti-looping factor and that its activity is required for neuronal remodeling.

The manuscript is clearly written, and the results are well organized. The observation that

knockdown of shep could increase the interaction frequency between the brat promoter and enhancer is interesting. However, the evidences are still weak to support key conclusions, and more mechanistic studies are needed to demonstrate causative links between Shep binding, enhancer-promoter interaction and chromatin status change. My specific points are listed below.

We thank the reviewer for their enthusiasm for our work. We have now performed many additional experiments in the appropriate matched cell types and obtained strong evidence that Shep directly associates with chromatin to regulate E-P looping and chromatin accessibility to control gene expression as a dedicated anti-looper.

Major points:

1. In Fig 1A, the RNA-seq tracks cannot show a clear change in expression of different brat transcripts. Experiments like isoform-specific RT-qPCR will be more quantitative.

Our new RT-qPCR data indicate clearly that all *brat* isoforms are upregulated in Shep-depleted BG3 cells and pupal neurons (Figure 1B and C).

2. The signal specificity of Shep ChIP-seq in the brat locus should be validated by ChIP-qPCR in control and shep-kd cells.

We added the requested validation in Figure S1.

3. In the lower panel of fig1A, it is not clear how the authors determine region 1, 2 and 3 that had significant change. Just by looking at the tracks, many other regions also changed signal. A more quantitative way with statistics to show the 4c-seq differential between control and shep KD is needed.

We added bar plots in Figure 1E to show sequencing depth signals at specific 4C-seq regions including display of statistical values where $p < 0.01$. Of the entire locus, only these three regions meet this criterion.

4. Data presented in fig2 simply indicates that the region 1 fragment works as an enhancer in BG3 cells. This is a typical enhancer reporter assay, and the enhancer activity is not dependent on Shep. I do not see clear evidence showing “Shep inhibits brat promoter looping with a neural enhancer” as stated in the title of fig2. Moreover, a reporter that contains a brat-unrelated but active enhancer in BG3 cells and the brat promoter should be used as a control to demonstrate specific interaction between the brat promoter and enhancer 1.

We have revised the title of Figure 2 to “Region 1 is a neural-specific enhancer, and Shep does not directly repress *brat* enhancer or promoter activities.”

To address the reviewer’s concern, we used published BG3 STARR-seq data (Yáñez-Cuna et al., *Genome Research*, 2014) to select an active BG3 enhancer unrelated to *brat*, which is located near the *dmGlut* gene. We cloned the *dmGlut* enhancer juxtaposed to the *brat-F* promoter in order to perform luciferase reporter assays and found that this enhancer also

increases luciferase expression driven by the *brat-F* promoter. However, this unrelated enhancer activates mildly, to an extent weaker than driven by the region #1 enhancer (see embedded Figure B). This result suggests that the *brat-F* promoter can be activated by other enhancers (when juxtaposed), but that region #1 has a stronger effect and may indeed function more specifically than an unrelated enhancer. We did not add these results to the revised manuscript because doing so would interrupt the flow of the results section.

Figure B. Region 1 activates *brat* promoter-driven luciferase expression more strongly than the *dmGlut* enhancer.

5. In fig.3, the authors showed shep-KD decreased the interaction frequency between the enhancer region 1 and the *brat-F* promoter. This does not necessarily mean that Shep inhibits enhancer-promoter looping, because similar effect could result from knocking down a typical repressor.

Our original data showed that Shep-KD leads to **increased** E-P looping. By performing the reporter assays, we already showed that Shep is not simply a direct transcriptional repressor of either the *brat-F* promoter or region #1 enhancer since luciferase expression driven by either of these elements is unchanged when Shep is knocked down (Figure 2E). In further support of this conclusion, our new *in vivo* GFP enhancer-trap data show that enhancer activities of regions #1 and #4 also function independently of Shep (Figure S4A-D). These data support the model that Shep does not act as a simple repressor of transcription activities *per se*, but instead may inhibit *brat* expression by antagonizing E-P looping.

6. In fig.3, the Shep ChIP-seq data is from BG3 cells, a cell line that originated from fly larval CNS. If Shep directly regulates *brat* in a stage-specific manner, the binding of Shep to the *brat* locus should be stage-specific: strong in the larval brain but weak or gone in the pupal brain. The authors can examine this by using ChIP-qPCR in control larva, control pupa and shep-KD pupa.

We agree with the reviewer and have made tremendous new efforts to profile chromatin association of Shep in both larval and pupal brains, but unfortunately we failed to generate adequate new brain ChIP-seq datasets of Shep, presumably due to reduced efficiency of aged antibodies (>10 years) in combination with the technical challenge of profiling dissected fly brains. We dissected ~1,000 brains for multiple rounds of ChIP-seq and ChIP-qPCR experiments using both larval and pupal brains, varying amounts of brains, crosslinking procedures (high/low PFA, dual crosslinking), and SDS concentrations. However, none of these trials generated robust enrichment for known chromatin targets of Shep, although these ChIP trials were simultaneously validated with successful positive controls for other factors in parallel. We also attempted anti-Shep CUT&RUN and CUT&Tag on native or crosslinked neurons, yet none of these trials succeeded either, despite parallel success with H3K4me1, H3K27ac, and H3K27me3 antibodies (described above). We agree that stage-specific Shep chromatin association could underlie stage-specific Shep functions, but unfortunately, these technical challenges prevented us from fully executing this thoughtful suggestion.

Instead, we replaced the BG3 Shep ChIP-seq profile with a Shep ChIP-seq profile from dissected wildtype larval brains that was generated more than 10 years ago by Dr. Leah Matzat, who also generated the original BG3 Shep ChIP-seq profile (Matzat et al., *PLoS Genetics*, 2012). This dataset was generated with identical protocols and antibodies to what we are currently using. Both Shep BG3 and larval brain ChIP-seq datasets display Shep chromatin association at the 3C/4C anchor region of the *brat-F* promoter (Figures 1 and 3). Shep chromatin association is also observed at the region #4 enhancer in Shep brain ChIP-seq data (Figure 3). These new results support our model that Shep directly inhibits enhancer accessibility and E-P looping to repress *brat* expression during neuronal remodeling.

7. In fig. 4A, the authors highlighted two regions with Shep-inhibited accessibility. These regions do not overlap with enhancer #1, the *brat-F* promoter or any Shep ChIP-seq peak. What are these regions, other enhancers?

As described in the response to Reviewer #1, our *in vivo* GFP reporter assays indicated that region #4 harbors enhancer activity in pupal neurons. The other region in the *brat* locus that is inhibited by Shep for accessibility in pupal neurons is marked by enhancer marks H3K4me1 and H3K27ac, suggesting that it may also serve as a neuronal enhancer (Figure 4A). We did not explore this region further.

The overall ATAC-seq signal in these two regions is low in pupal neurons even after shep-KD, compared to that of the enhancer and promoter regions, and also much lower than that of the same region at larval stage. And the increase upon shep-KD seems marginal. Similar mild change is shown in the *Myc* locus (Fig. S3A). These results indicate that the effect on chromatin accessibility by shep-KD could be indirect and mild.

These two regions show a statistically significant (FDR=0.07 and fold change=1.3) increase in chromatin accessibility upon Shep knockdown. In control larval neurons, higher accessibility of these sites correlates with higher *brat* expression. In control pupal neurons, lower accessibility of these sites correlates with lower *brat* expression (Figure 4A). This tight correlation of accessibility and gene expression throughout development suggests that these regions may serve as regulatory elements that help control dynamic *brat* expression.

Mild changes of accessibility upon Shep depletion may become apparent due to possibly transient changes of accessibility. These changes could then perhaps allow transient E-P interactions that could activate *brat* transcription.

Alternatively, mild changes of accessibility may result from changes that occur in only specific neurons within the total population.

In addition, changes in accessibility may not be linearly related to changes in looping frequency and transcription. Thus even mild changes of accessibility may lead to strong effects on expression.

Lastly, the RNAi-depleted neurons used for ATAC-seq assays have ~60% depletion of Shep (Chen et al., *Development*, 2018), which may also explain mild effects. Despite these mild

changes, Shep regulation of accessibility is prominent when examined genome-wide (Figure 4B-D). These accessibility changes are highly correlated with Shep-regulated gene expression changes (Figure 4E), suggesting direct Shep regulation. Some of these possible scenarios are now discussed on p. 11 of the Discussion.

Regarding the relative accessibility of nearby enhancer and promoter regions, it is well known that promoter regions have exceptionally high accessibility compared to other features across the genome (Thurman et al., *Nature*, 2014; Reddington et al., *Developmental Cell*, 2020). The region #1 enhancer likely also has high accessibility because it is located nearby the *I(2)37Cg* promoter, the gene encoded on the opposite strand.

In summary, our computational analyses demonstrated highly significant correlations among Shep chromatin binding, Shep-dependent enhancer accessibility, and Shep-dependent gene expression (Figure 4E and text on p. 9). These results support a model in which Shep directly antagonizes *brat* enhancer accessibility, E-P looping, and gene expression.

I don't see clear evidence that supports the conclusion in Page 5 "..., indicating correspondence between increased chromatin accessibility and increased looping frequency in Shep-depleted neurons".

In the original manuscript, we identified increased E-P looping between the *brat-F* promoter and region #1 enhancer in Shep-depleted pupal brains. A neighboring region #4 displayed elevated accessibility in Shep-depleted neurons. In the revised manuscript, we performed *in vivo* GFP reporter assays to confirm enhancer activities of region #4 and conduct additional Taqman 3C-qPCR that showed increased E-P looping between *brat-F* promoter and enhancer region #4 in Shep-depleted pupal brains. This result demonstrates correspondence between increased enhancer accessibility and increased E-P looping of region #4 enhancer to *brat-F* promoter upon Shep depletion.

8. H3K27Ac gives better enhancer prediction than H3K4me1 do. Have the authors used the H3K27Ac-predicted enhancers for the correlation study in Fig. 4B-E?

It has been shown that H3K27ac modification labels active enhancers (Creyghton et al., *PNAS*, 2010) while H3K4me1 modification labels all enhancers (Heintzman et al., *Nature*, 2009). Since we did not want to limit our study only to active enhancers at a given stage, we feel strongly that H3K4me1 is a better marker for this purpose.

9. In fig. 4B-E, it is hard to tell how many of the Shep-inhibited and promoted sites are directly regulated by Shep. Have the authors incorporate Shep ChIP-seq data in the analyses, for example, to see whether only the Shep-bound genes changed accessibility?

As explained above, we have integrated new Shep ChIP-seq data from larval brain with ATAC-seq, CUT&Tag, and RNA-seq data in larval and pupal neurons in our revised manuscript. We found that chromatin association of Shep is significantly enriched ($p=1.7e-8$, odds ratio=4.5) at 32% of genes (26 of 82) that display Shep-dependent inhibition of both gene expression and

enhancer accessibility (Figure 4E and text on p. 9).

Minor points:

1. In fig. 1A, the 4C anchor point seems on the Shep ChIP-seq peak but not on the brat-F promoter. It is hard to tell the distance between the 4C anchor and the brat-F promoter with the presented scale.

The anchor is located 805 bp from the transcription start site of *brat-F*. We have updated the scale bar to better reflect the distance. We also added a diagram to Figure 2A to show the exact fragments used for luciferase assays.

2. In fig. 3, the 3C/4C anchor (covering almost two ChIP-seq peaks) looks much broader than the 4C anchor (covering a half peak) shown in fig. 1. Does this mean the 3C anchor is a much broader region?

Yes, the 4C anchor is nested within the 3C anchor because there are two digestions to generate fragments for 4C-seq whereas 3C-qPCR templates are generated after one digestion. The 4C anchor is 484 bp, and the 3C anchor is 2,352 bp. We have updated Figures 3A and 4A by labeling only the 3C anchor to improve clarity.

Reviewer #3:

Previously, the Lei lab have shown that the conserved RNA-binding protein Shep inhibits brat expression during neuronal remodelling in the *Drosophila* pupal brain. Here, Chen and colleagues describe their novel findings, where they propose that Shep inhibits brat expression by antagonising the long-range interaction of the brat isoform F (brat-F) promoter with a newly identified enhancer.

Key points to address:

i) The Shep-dependent looping from enhancer 1 to the Brat-F promoter data is both convincing and an exciting finding. However, it is not clear to what degree this impacts on brat-F expression, and the expression of other isoforms. brat-A/E, brat-B/C and brat-F all contain Shep ChIP-seq peaks just upstream of their promoters, raising the possibility that Shep could bind to these upstream regions to directly inhibit transcription. There is some indication that this could be happening from the RNA-seq data shown in Fig.1, as it appears that the longer isoforms (brat-E, and possibly brat-C) show increased expression upon Shep KD. If Shep is only regulating expression via inhibiting looping to the brat-F promoter, then levels of brat-E/C would be unaffected, raising the possibility that Shep is either regulating E-P looping to brat-E/C, and/or it is acting directly to repress transcription. The *in vitro* experiments with the brat-F promoter (which contains a Shep binding site) do at least seem to argue against direct repression by Shep, however, it is not known for the other isoforms. Testing the activity of the brat-A/E and brat-B/C promoters in a Shep-KD condition, *in vitro*, could help rule this out. qPCR of brat-F (using the unique first exon) might also help.

We thank Dr. Southall for his enthusiasm for our work. We previously focused on *brat-F* because it is the dominantly expressed isoform in pupal neurons. Meanwhile, we did not propose exclusive, *brat-F*-specific Shep-dependent inhibition of expression and E-P looping. To interrogate additional isoforms, we performed additional RT-qPCR and found that Shep inhibits expression of all *brat* isoforms in BG3 cells and pupal neurons (Figure 1A-C). We also performed additional *ex vivo* luciferase assays to test *brat-A/E* and *brat-B/C* promoter activities in control and Shep-depleted BG3 cells. Like *brat-F*, both sets of promoters drive expression of luciferase but are unaffected by Shep knockdown (Figure 2F and G). These results confirm that Shep does not directly inhibit activity of any *brat* promoters in this context.

To explore whether Shep regulates E-P looping of the *brat-B/C* promoter, we also performed additional Taqman 3C-qPCR in pupal brains to quantify interactions between the *brat-B/C* promoter and regions #1 and #4, as well as one more flanking region on the 3' side. The *brat-A/E* promoter was not tested since it is very close to regions 1 and 4, so 3C templates will be dominated by proximal ligation events rather than three-dimensional looping. In control pupal brains, we observed low interactions (note different signal scales between Figure 3A and Figure S3) between the *brat-B/C* promoter and all three loci, which correlates with overall low expression of *brat-B/C* isoforms in pupal brains (Figure S3). Upon Shep depletion, interaction between the *brat-B/C* promoter and only the 3' region are marginally significantly increased (Figure S3). These results suggest perhaps similar Shep-dependent regulation of chromatin looping to *brat-B/C* and *brat-F* isoform promoters. For simplicity, we decided to focus on *brat-F*, which is the dominant isoform expressed during neuronal remodeling and displays high levels of E-P looping, allowing for easier analysis.

ii) How can the authors explain why the changes in chromatin accessibility are observed at regions neighbouring the identified regulatory regions, rather than at those regions themselves (Fig 4)?

Our new data shows that the nearby region #4 is also an enhancer. This conclusion was derived by testing its ability to drive expression *in vivo* by cloning region #4 just upstream of the *hsp70* promoter driving a GFP reporter. In this assay, region #4 is observed to act as an enhancer specifically in pupal brains (Figure 3G). Furthermore, region #4 also loops to the *brat-F* promoter *in vivo*, and E-P looping increases when *shep* is knocked down (Figure 3A). Although not originally identified by our 4C-seq assay in BG3 cells, which are derived from the larval nervous system, region #4 may actually be the key regulatory region involved in *in vivo* regulation of *brat* expression during neuronal remodeling.

Minor points:

i) It would be helpful if the *brat-F* promoter region in the *in vitro* experiments was clearly defined in the main text and in the corresponding Fig 2A. Annotation of the *brat-F* promoter region on the figure would be helpful to provide context (i.e. relative position to enhancers and whether it contains a Shep binding site).

We have revised Figure 2A to clearly indicate the location of the cloned *brat-F* promoter, and a bar has been added to reflect that information. We have also added the *brat-A/E* and *brat-B/C* promoters.

ii) Figures 4B and C are very confusing and not easily interpreted.

We have removed original Figure 4B and C by integrating relevant information into new Figure 4E and the main text on p. 9.

iii) There is good evidence that *brat* regulation is important for neuronal remodelling, however, it is less clear for general neuronal maturation. I suggest that the text is re-worded to only mention remodelling, in order to prevent misinterpretation/confusion.

We have revised the text on pp. 1-2 and 9-10 to emphasize neuronal remodeling instead of neuronal maturation.

REVIEWER COMMENTS

Reviewer #1 (Remarks to the Author):

Prior to resubmission of the revised manuscript, my major concerns were the following:

1. There was insufficient evidence that a region contacting brat-F promoter more frequently upon Shep-KD (knock-down) was relevant for brat-F upregulation in pupal neurons.
2. The data did not support the main conclusion of the study that Shep is an "enhancer-promoter anti-looping factor".

Point 1 is now nicely addressed because the authors show through substantial new work that region #4 interacting more frequently with brat-F promoter upon Shep-KD is a pupal brain enhancer (new Fig. 3B-G), and that deleting this region slightly decreases brat-F expression in pupal brains (new Fig. 3H). These results make sense to me.

Point 2 was not addressed, and I remain as confused as before by the extreme focus of the manuscript on what I find to be a misleading buzzword ("anti-looper"). The notion that Shep acts by somehow preventing looping of brat-F to a distal enhancer is communicated in the title ("anti-looping factor"), the keywords ("anti-looper"), the abstract ("Our results provide evidence for a dedicated chromatin anti-looping factor"), several results subsection titles ("Shep inhibits brat promoter looping (...)", "Shep inhibits neural brat promoter-enhancer looping") and the discussion ("Our results report a novel, dedicated chromatin anti-looping factor"...). The basis of this claim is that in Shep-KD, brat-F promoter interacts more with a distal enhancer, and that enhancer activity is unaffected by Shep depletion in transgenic reporter assays in brains that do not quantify the activity of the endogenous enhancer in situ (this would be challenging to do). This evidence is therefore indirect, and the alternative hypothesis that the subtle (1.4-fold) increase in brat expression upon Shep-KD is explained by increased chromatin accessibility of pupal enhancer #4 in Shep-KD (Fig. 4A) is more obvious to me. The authors also clearly agree that Shep is implicated in closing brat distal enhancers in pupal neurons to reduce brat transcription (they write in the rebuttal: "These findings are consistent with the possibility that Shep-mediated enhancer closure during remodeling is related to the underlying mechanism behind Shep-dependent temporal regulation of gene expression").

My other points were nicely addressed.

In summary, I find the data showing that chromatin accessibility is strikingly remodelled between larval and pupal neurons, and that Shep-KD impairs "closing" of many larval enhancers in pupal neurons and upregulates linked genes sufficiently interesting in itself. I am not convinced that Shep is an anti-looping factor and I would recommend strongly toning this interpretation down throughout the manuscript.

Reviewer #2 (Remarks to the Author):

In the revised version of the manuscript, Chen et al. have added a substantial amount of new data in an effort to address the reviewers' comments. Although the data is now more comprehensive, I find there still lacks a strong and direct evidence to support their main conclusion, that Shep antagonizes E-P looping to inhibit brat expression. In particular, my main concerns are

1. the authors examined enhancer activity of two genomic regions that show enhanced looping frequency in response to Shep depletion (region 1 in luciferase assay and region 1 and 4 in in vivo GFP reporter). Their result showed that the enhancer activity is not dependent on Shep because Shep RNAi did not change enhancer activity. The authors claimed, "Shep cannot directly repress the activity of either enhancer", and "Shep functions as an antagonist of E-P looping in order to negatively regulate

brat transcription". I agree with the first statement, while I find the second claim overstated, because Shep may affect brat expression in other ways (for example, by affecting brat RNA stability or through another factor that regulates brat transcripts).

2. in fig 3, the authors show that both region 1 and 4 have high interaction frequency with the brat-F promoter in the pupal brain but not in the larval brain. The frequency is further increased upon Shep RNAi. However, in the GFP reporter test, region 1 drives GFP expression in the larval brain while region 4 in the pupal brain. Does this mean stage-specific E-P looping is neither sufficient nor necessary for gene expression (or at least for enhancer activity)?

3. in fig 4, region1 and region 4 both have high chromatin accessibility in the larval stage and only region 4 becomes closed in pupal neurons. Again when combined with the GFP reporter test, does this result indicate chromatin accessibility is not correlated with enhancer activity, as region 1 is still open but not acts as an enhancer in pupa while region 4 is more open in larva but only active in pupa?

4. Based on fig 3 and S3, Shep affects the interaction of region 1 and 4 with brat-F but not with other brat promoters. However, all isoforms became up-regulated (fig 1C-D). Does this mean that the regulation by Shep on brat transcription is neither isoform-specific nor stage-specific? If so, what is the function of stage-specific E-P looping?

Reviewer #3 (Remarks to the Author):

The authors have performed an impressive set of extra experiments to address the concerns raised and this is a much improved manuscript. I recommend publication without further revision.

REVIEWER COMMENTS

Reviewer #1:

Prior to resubmission of the revised manuscript, my major concerns were the following:

1. There was insufficient evidence that a region contacting brat-F promoter more frequently upon Shep-KD (knock-down) was relevant for brat-F upregulation in pupal neurons.
2. The data did not support the main conclusion of the study that Shep is an “enhancer-promoter anti-looping factor”.

Point 1 is now nicely addressed because the authors show through substantial new work that region #4 interacting more frequently with brat-F promoter upon Shep-KD is a pupal brain enhancer (new Fig. 3B-G), and that deleting this region slightly decreases brat-F expression in pupal brains (new Fig. 3H). These results make sense to me.

We appreciate the reviewer's enthusiasm for our work.

Point 2 was not addressed, and I remain as confused as before by the extreme focus of the manuscript on what I find to be a misleading buzzword (“anti-looper”). The notion that Shep acts by somehow preventing looping of brat-F to a distal enhancer is communicated in the title (“anti-looping factor”), the keywords (“anti-looper”), the abstract (“Our results provide evidence for a dedicated chromatin anti-looping factor”), several results subsection titles (“Shep inhibits brat promoter looping (...)”, “Shep inhibits neural brat promoter-enhancer looping”) and the discussion (“Our results report a novel, dedicated chromatin anti-looping factor”...). The basis of this claim is that in Shep-KD, brat-F promoter interacts more with a distal enhancer, and that enhancer activity is unaffected by Shep depletion in transgenic reporter assays in brains that do not quantify the activity of the endogenous enhancer in situ (this would be challenging to do). This evidence is therefore indirect, and the alternative hypothesis that the subtle (1.4-fold) increase in brat expression upon Shep-KD is explained by increased chromatin accessibility of pupal enhancer #4 in Shep-KD (Fig. 4A) is more obvious to me. The authors also clearly agree that Shep is implicated in closing brat distal enhancers in pupal neurons to reduce brat transcription (they write in the rebuttal: “These findings are consistent with the possibility that

Shep-mediated enhancer closure during remodeling is related to the underlying mechanism behind Shep-dependent temporal regulation of gene expression”).

We have revised the title to “Temporal inhibition of chromatin looping and enhancer accessibility during neuronal remodeling” and toned down the conclusions in the abstract in two separate sentences on p. 2 and in the Discussion on p.10 (see tracked changes).

My other points were nicely addressed.

In summary, I find the data showing that chromatin accessibility is strikingly remodelled between larval and pupal neurons, and that Shep-KD impairs “closing” of many larval enhancers in pupal neurons and upregulates linked genes sufficiently interesting in itself. I am not convinced that Shep is an anti-looping factor and I would recommend strongly toning this interpretation down throughout the manuscript.

We thank this reviewer for their enthusiasm for our work. We have toned down our interpretations as indicated above.

Reviewer #2:

In the revised version of the manuscript, Chen et al. have added a substantial amount of new data in an effort to address the reviewers’ comments. Although the data is now more comprehensive, I find there still lacks a strong and direct evidence to support their main conclusion, that Shep antagonizes E-P looping to inhibit *brat* expression. In particular, my main concerns are

1. the authors examined enhancer activity of two genomic regions that show enhanced looping frequency in response to Shep depletion (region 1 in luciferase assay and region 1 and 4 in in vivo GFP reporter). Their result showed that the enhancer activity is not dependent on Shep because Shep RNAi did not change enhancer activity. The authors claimed, “Shep cannot directly repress the activity of either enhancer”, and “Shep functions as an antagonist of E-P looping in order to negatively regulate *brat* transcription” . I agree with the first statement, while I find the second claim overstated, because Shep may affect *brat* expression in other ways (for example, by affecting *brat* RNA stability or through another factor that regulates *brat* transcripts).

We appreciate the reviewer’s enthusiasm for our work. As detailed above, we have revised the title, toned down the conclusions, and reduced use of the “anti-looping” terminology. To address the reviewer’s concern, we added a statement to the Discussion on p. 11 “Our results do not rule out the possibility that Shep also downregulates *brat* or other genes by posttranscriptional mechanisms.”

2. in fig 3, the authors show that both region 1 and 4 have high interaction frequency with the *brat*-F promoter in the pupal brain but not in the larval brain. The frequency is further increased upon Shep RNAi. However, in the GFP reporter test, region 1 drives GFP expression in the

larval brain while region 4 in the pupal brain. Does this mean stage-specific E-P looping is neither sufficient nor necessary for gene expression (or at least for enhancer activity)?

As we previously mentioned on p. 12, we and others previously observed that E-P looping is not necessarily sufficient for gene expression (Ghavi-Helm et al., *Nature Genetics*, 2019).

With respect to enhancer region #1, we did not examine its looping to the *brat-B/C* promoter in larval brains so cannot address the reviewer's question about necessity. Furthermore, region #1 is juxtaposed to the *brat-A/E* promoter so likely does not require looping in order to activate this promoter.

3. in fig 4, region1 and region 4 both have high chromatin accessibility in the larval stage and only region 4 becomes closed in pupal neurons. Again when combined with the GFP reporter test, does this result indicate chromatin accessibility is not correlated with enhancer activity, as region 1 is still open but not acts as an enhancer in pupa while region 4 is more open in larva but only active in pupa?

The reviewer is correct that there is no correlation between enhancer activity and accessibility according to our observations and those of many others (Arnold et al., *Science*, 2013; Shashikant et al., *BMC Genomics*, 2018; reviewed in Bozek and Gompel, *BioEssays*, 2020). However, it is generally believed that changes in chromatin accessibility of enhancers likely reflect changes in chromatin occupancy of relevant bound factors, such as transcription factors, that control gene expression.

4. Based on fig 3 and S3, Shep affects the interaction of region 1 and 4 with *brat-F* but not with other *brat* promoters. However, all isoforms became up-regulated (fig 1C-D). Does this mean that the regulation by Shep on *brat* transcription is neither isoform-specific nor stage-specific? If so, what is the function of stage-specific E-P looping?

All *brat* isoforms are upregulated upon Shep depletion in figure 1B-C, so Shep inhibition of *brat* expression is not isoform-specific. We also showed that Shep regulates *brat* expression specifically in pupal but not larval neurons, indicating stage-specific regulation (Figures 1A and Chen et al, *Development*, 2018).

As this reviewer mentioned, we showed that Shep specifically inhibits E-P looping of the *brat-F* isoform, indicating isoform-specific regulation of E-P looping. We also showed that stage-specific E-P looping correlates with proper expression of *brat-F*, which is the major isoform in pupal neurons.

We did not fully interrogate stage-specific E-P looping of other *brat* isoforms nor did we identify all *brat* locus enhancers. Because of the close proximity of enhancer regions #1 and #4 to the *brat-A/E* promoter, E-P looping is unlikely to be relevant in this case. It is also possible that Shep regulation of other *brat* isoforms may be achieved through other mechanisms, which are not the focus of this manuscript. As previously mentioned, we added a statement on p. 11 discussing the possibility that Shep may also regulate *brat* or other genes via post-transcriptional mechanisms.

Reviewer #3:

The authors have performed an impressive set of extra experiments to address the concerns raised and this is a much improved manuscript. I recommend publication without further revision.

We thank the reviewer for their enthusiasm for our work.